# Distributed Multitask Reinforcement Learning with Quadratic Convergence

**Rasul Tutunov**
PROWLER.io
Cambridge, United Kingdom
rasul@prowler.io

**Dongho Kim**
PROWLER.io
Cambridge, United Kingdom
dongho@prowler.io

**Haitham Bou-Ammar**
PROWLER.io
Cambridge, United Kingdom
haitham@prowler.io

## Abstract

Multitask reinforcement learning (MTRL) suffers from scalability issues when the number of tasks or trajectories grows large. The main reason behind this drawback is the reliance on centralised solutions. Recent methods exploited the connection between MTRL and general consensus to propose scalable solutions. These methods, however, suffer from two drawbacks. First, they rely on predefined objectives, and, second, exhibit linear convergence guarantees. In this paper, we improve over state-of-the-art by deriving multitask reinforcement learning from a variational inference perspective. We then propose a novel distributed solver for MTRL with *quadratic convergence guarantees*.

## 1 Introduction

Reinforcement learning (RL) allows agents to solve sequential decision-making problems with limited feedback. Applications with these characteristics are ubiquitous ranging from stock-trading [1] to robotics control [2, 3]. Though successful, RL methods typically require substantial amounts of data and computation for successful behaviour. Multitask and transfer learning [4–6, 2, 7] techniques have been developed to remedy these problems by allowing for knowledge reuse between tasks to bias initial behaviour. Unfortunately, such methods suffer from scalability constraints when the number of tasks or policy dimensions grows large.

Two promising directions remedy these scalability problems. In the first, tasks are streamed online and models are fit iteratively. Such an alternative has been well-explored under the name of lifelong RL [8, 9]. When considering lifelong learning, however, one comes to recognise that these improvements in computation come hand-in-hand with a decrease in the model's accuracy due to the usage of approximations to the original loss (e.g., second-order expansions [10]), as well as the unavailability of all tasks in batch. Interested readers are referred to [11] for an in-depth discussion of the limitations of lifelong reinforcement learners.

The other direction based on decentralised optimisation remedies scalability and accuracy constraints by distributing computation across multiple units. Though successful in supervised learning [12], this direction is still to be well-explored in the context of MTRL. Recently, however, the authors in [11] proposed a distributed solver for MTRL with linear convergence guarantees based on the Alternating Direction Method of Multipliers (ADMM). Their method relied on a connection between MTRL and distributed general consensus. However, such ADMM-based techniques suffer from the

following drawbacks. First, these algorithms only achieve linear convergence in the order of $\mathcal{O}\left(1/k\right)$ with $k$ being the iteration count. Second, for linear convergence additional restrictive assumptions on the penalty terms have to be imposed. Finally, they require large number of iterations to arrive at accurate (in terms of consensus error) solutions as noted by [13] and validated in our experiments, see Section 5.

In this paper, we remedy the above problems by proposing a distributed solver for MTRL that exhibits *quadratic convergence*. Contrary to [11], our technique does not impose restrictive assumptions on the reinforcement learning loss function and can thus be deemed more general. We achieve our results in two-steps. First, we reformulate MTRL as variational inference. Second, we map the resultant objective to general consensus that allows us to exploit the symmetric and diagonal dominance property of the curvature of our dual problem. We show our novel distributed solver using Chebyshev polynomials has quadratic convergence guarantees.

We analyse the performance of our method both theoretically and empirically. On the theory side, we formally prove quadratic convergence. On the empirical side, we show that our new technique outperforms state-of-the-art methods from both distributed optimisation and lifelong reinforcement learning on a variety of graph topologies. We further show that these improvements arrive at relatively small increases in the communication overhead between the nodes.

## 2   Background

Reinforcement learning (RL) [14] algorithms are successful in solving sequential decision making (SDM) tasks. In RL, the agent's goal is to sequentially select actions that maximise its total expected return. We formalise such problems as a Markov decision process (MDP) $\mathcal{Z} = \langle \mathcal{X}, \mathcal{A}, \mathcal{P}, \mathcal{R}, \gamma \rangle$ where $\mathcal{X} \subseteq \mathbb{R}^d$ is the set of states, $\mathcal{A} \subseteq \mathbb{R}^m$ is the set of possible actions, $\mathcal{P} : \mathcal{X} \times \mathcal{A} \times \mathcal{X} \mapsto [0,1]$ represents the state transition probability describing the task's dynamics, $\mathcal{R} : \mathcal{X} \times \mathcal{A} \times \mathcal{X} \mapsto \mathbb{R}$ is the reward function measuring the agent's performance, and $\gamma \in [0,1)$ is the discount factor. The dynamics of an RL problem commence as follows: at each time step $h$, the agent is at state $\boldsymbol{x}_h \in \mathcal{X}$ and has to choose an action $\boldsymbol{a}_h \in \mathcal{A}$ transitioning it to a new state $\boldsymbol{x}_{h+1} \sim p(\boldsymbol{x}_{h+1}|\boldsymbol{x}_h, \boldsymbol{a}_h)$ as given by $\mathcal{P}$. This transition yields a reward $r_{h+1} = \mathcal{R}(\boldsymbol{x}_h, \boldsymbol{a}_h, \boldsymbol{x}_{h+1})$. We assume that actions are generated by a policy $\pi : \mathcal{X} \times \mathcal{A} \mapsto [0,1]$, which is defined as a distribution over state-action pairs, i.e., $\pi(\boldsymbol{a}_h|\boldsymbol{x}_h)$ is the probability of choosing action $\boldsymbol{a}_h$ in a state $\boldsymbol{x}_h$. The goal of the agent is to find an optimal policy $\pi^*$ that maximises its expected return given by: $\mathbb{E}_\pi\left[\sum_{h=1}^{H}\gamma^h r_h\right]$, with $H$ being the horizon length.

**Policy Search RL** parameterises a policy by a vector of unknown parameters $\boldsymbol{\theta}$. As such, the RL problem is transformed to a one of searching over the parameter space for $\boldsymbol{\theta}^\star$ that maximises:

$$\mathcal{J}(\boldsymbol{\theta}) = \mathbb{E}_{p_{\boldsymbol{\theta}}(\boldsymbol{\tau})}[\mathfrak{R}(\boldsymbol{\tau})] = \int_{\boldsymbol{\tau}} p_{\boldsymbol{\theta}}(\boldsymbol{\tau})\mathfrak{R}(\boldsymbol{\tau})d\boldsymbol{\tau}, \tag{1}$$

where a trajectory $\boldsymbol{\tau}$ is a sequence of accumulated state-action pairs $[\boldsymbol{x}_{0:H}, \boldsymbol{a}_{0:H}]$. Furthermore, the probability of acquiring a certain trajectory, $p_{\boldsymbol{\theta}}(\boldsymbol{\tau})$, and the total reward $\mathfrak{R}(\boldsymbol{\tau})$ for a trace $\boldsymbol{\tau}$ are defined as: $p_{\boldsymbol{\theta}}(\boldsymbol{\tau}) = \mathcal{P}_0(\boldsymbol{x_0}) \prod_{h=1}^{H} p(\boldsymbol{x}_{h+1}|\boldsymbol{x}_h, \boldsymbol{a}_h)\pi_{\boldsymbol{\theta}}(\boldsymbol{a}_h|\boldsymbol{x}_h)$, and $\mathfrak{R}(\boldsymbol{\tau}) = \frac{1}{H}\sum_{h=0}^{H} r_{h+1}$, with $\mathcal{P}_0 : \mathcal{X} \mapsto [0,1]$ being the initial state distribution.

Policy search can also be cast **as variational inference** by connecting RL and probabilistic inference [15–18]. In this formulation the goal is to derive the posterior distribution over trajectories conditioned on a desired output, given a prior trajectory distribution. The desired output is denoted as a binary random variable $\hat{\mathcal{R}}$, where $\hat{\mathcal{R}} = 1$ indicates the optimal reward event. This is typically related to trajectory rewards using $p(\hat{\mathcal{R}} = 1|\boldsymbol{\tau}) \propto \exp(\mathfrak{R}(\boldsymbol{\tau}))$. With this definition, the optimisation objective $\mathcal{J}(\boldsymbol{\theta})$ in Equation 1 is $p_{\boldsymbol{\theta}}(\hat{\mathcal{R}} = 1) = \int_{\boldsymbol{\tau}} p(\hat{\mathcal{R}} = 1|\boldsymbol{\tau})p_{\boldsymbol{\theta}}(\boldsymbol{\tau})d\boldsymbol{\tau}$. From the log-marginal of the binary event, we can write the evidence lower bound (ELBO). The ELBO is derived by introducing a variational distribution $q_{\boldsymbol{\phi}}(\boldsymbol{\tau})$ and applying Jensen's inequality:

$$\log p_{\boldsymbol{\theta}}(\hat{\mathcal{R}}) \geq \int_{\boldsymbol{\tau}} q_{\boldsymbol{\phi}}(\boldsymbol{\tau})\left[\log p(\hat{\mathcal{R}}|\boldsymbol{\tau}) + \log \frac{p_{\boldsymbol{\theta}}(\boldsymbol{\tau})}{q_{\boldsymbol{\phi}}(\boldsymbol{\tau})}\right]d\boldsymbol{\tau} = \mathbb{E}_{q_{\boldsymbol{\phi}}(\boldsymbol{\tau})}\left[\log p(\hat{\mathcal{R}}|\boldsymbol{\tau})\right] - \mathcal{D}_{\mathrm{KL}}(q_{\boldsymbol{\phi}}(\boldsymbol{\tau})\|p_{\boldsymbol{\theta}}(\boldsymbol{\tau})),$$

with $\mathcal{D}_{\mathrm{KL}}\left(q(\boldsymbol{\tau})\|p(\boldsymbol{\tau}))\right)$ being the Kullback Leibler divergence between $q(\boldsymbol{\tau})$ and $p(\boldsymbol{\tau})$.

# 3 Multitask Reinforcement Learning as Variational Inference

RL algorithms require substantial amounts of trajectories and learning times for successful behaviour. Acquiring large training samples easily leads to wear and tear on the system and thus worsens the problem. When data is scarce, learning task policies jointly through *multi-task reinforcement learning* (MTRL) rather than independently significantly improves performance [4, 19]. In MTRL, the agent is faced with a series of $T$ SDM tasks $\mathcal{Z}^{(1)}, ..., \mathcal{Z}^{(T)}$. Each task is an MDP denoted by $\mathcal{Z}^{(t)} = \langle \mathcal{X}^{(t)}, \mathcal{A}^{(t)}, \mathcal{P}^{(t)}, \mathcal{R}^{(t)}, \gamma^{(t)} \rangle$, and the goal for the agent is to learn a set of optimal policies $\mathbf{\Pi}^{\star} = \{\pi_{\boldsymbol{\theta}^{(1)}}^{\star}, ..., \pi_{\boldsymbol{\theta}^{(T)}}^{\star}\}$ with corresponding parameters $\boldsymbol{\Theta}^{\star} = \{\boldsymbol{\theta}^{(1)\star}, ..., \boldsymbol{\theta}^{(T)\star}\}$. Rather than defining the optimisation objective directly as done in [10, 4], we provide a probabilistic modeling view of the problem by framing MTRL as an instance of variational inference. We define a set of reward binary events $\hat{\mathcal{R}}_1, \ldots, \hat{\mathcal{R}}_T \in \{0, 1\}^T$ where $p(\hat{\mathcal{R}}_k | \boldsymbol{\tau}_k) \propto \exp\left(\mathfrak{R}_k(\boldsymbol{\tau}_k)\right)$. Here, trajectories are assumed to be latent, and the goal of the agent is to determine a set of policy parameters that assign high density to trajectories with high rewards. In other words, the goal is to find a set of policies that maximise the log-marginal of the reward events:

$$\log p_{\boldsymbol{\theta}_1:\boldsymbol{\theta}_T}\left(\hat{\mathcal{R}}_1, \ldots \hat{\mathcal{R}}_T\right) = \log \int_{\boldsymbol{\tau}_1} \cdots \int_{\boldsymbol{\tau}_T} \prod_{t=1}^{T} p(\hat{\mathcal{R}}_t | \boldsymbol{\tau}_t) p_{\boldsymbol{\theta}_t}(\boldsymbol{\tau}_t) d\boldsymbol{\tau}_1 \ldots d\boldsymbol{\tau}_T,$$

where $p_{\boldsymbol{\theta}_t}(\boldsymbol{\tau}_t)$ is the trajectory density for task $t$: $p_{\boldsymbol{\theta}_t}(\boldsymbol{\tau}_t) = \mathcal{P}_0^{(t)}(\boldsymbol{x}_0) \prod_{h=1}^{H_t} p^{(t)}\left(\boldsymbol{x}_{h+1}^{(t)} | \boldsymbol{x}_h^{(t)}, \boldsymbol{a}_h^{(t)}\right) \pi_{\boldsymbol{\theta}_t}\left(\boldsymbol{a}_h^{(t)} | \boldsymbol{x}_h^{(t)}\right)$. To handle the intractability in computing the above integrals, we derive an ELBO using a variational distribution $q_{\boldsymbol{\phi}}(\boldsymbol{\tau}_1, \ldots, \boldsymbol{\tau}_T)$:

$$\log \int_{\boldsymbol{\tau}_1} \cdots \int_{\boldsymbol{\tau}_T} \prod_{t=1}^{T} p(\hat{\mathcal{R}}_t | \boldsymbol{\tau}_t) p_{\boldsymbol{\theta}_t}(\boldsymbol{\tau}_t) d\boldsymbol{\tau}_1 \ldots d\boldsymbol{\tau}_T \geq \mathbb{E}_{q_{\boldsymbol{\phi}}(\cdot)}\left[\sum_{t=1}^{T} \log p\left(\hat{\mathcal{R}}_t | \boldsymbol{\tau}_t\right) + \log \frac{\prod_{t=1}^{T} p_{\boldsymbol{\theta}_t}(\boldsymbol{\tau})}{q_{\boldsymbol{\phi}}(\cdot)}\right]$$

Using the above, the optimisation objective of multitask reinforcement learning can be written as:

$$\max_{\boldsymbol{\phi}, \boldsymbol{\theta}_1:\boldsymbol{\theta}_T} \mathbb{E}_{q_{\boldsymbol{\phi}}(\boldsymbol{\tau}_1,...,\boldsymbol{\tau}_T)}\left[\sum_{t=1}^{T} \log p\left(\hat{\mathcal{R}}_t | \boldsymbol{\tau}_t\right)\right] + \mathbb{E}_{q_{\boldsymbol{\phi}}(\boldsymbol{\tau}_1,...,\boldsymbol{\tau}_T)}\left[\log \frac{\prod_{t=1}^{T} p_{\boldsymbol{\theta}_t}(\boldsymbol{\tau})}{q_{\boldsymbol{\phi}}(\boldsymbol{\tau}_1, \ldots, \boldsymbol{\tau}_T)}\right].$$

We assume a mean-field variational approximation [20], i.e., $q_{\boldsymbol{\phi}}(\boldsymbol{\tau}_1, \ldots, \boldsymbol{\tau}_T) = \prod_{t=1}^{T} q_{\boldsymbol{\phi}_t}(\boldsymbol{\tau}_t)$. Furthermore, we assume that the distribution[1] $q_{\boldsymbol{\phi}_t}(\boldsymbol{\tau}_t)$ follows that of $p_{\boldsymbol{\theta}_t}(\boldsymbol{\tau}_t)$. Hence, we write:

$$\max_{\boldsymbol{\phi}_1:\boldsymbol{\phi}_T, \boldsymbol{\theta}_1:\boldsymbol{\theta}_T} \sum_{t=1}^{T} \mathbb{E}_{q_{\boldsymbol{\phi}_t}(\boldsymbol{\tau}_t)}\left[\log p\left(\hat{\mathcal{R}}_t | \boldsymbol{\tau}_t\right)\right] - \sum_{t=1}^{T} \mathcal{D}_{\text{KL}}\left(q_{\boldsymbol{\phi}_t}(\boldsymbol{\tau}_t) || p_{\boldsymbol{\theta}_t}(\boldsymbol{\tau}_t)\right). \tag{2}$$

So far, we discussed MTRL assuming independence between policy parameters $\boldsymbol{\theta}_1, \ldots, \boldsymbol{\theta}_T$. To benefit from shared knowledge between tasks, we next introduce *coupling* by allowing for parameter sharing across MDPs. Inspired by stochastic variational inference [21], we decompose $\boldsymbol{\theta}_t = \boldsymbol{\Theta}_{\text{sh}} \tilde{\boldsymbol{\theta}}_t$, where $\boldsymbol{\Theta}_{\text{sh}}$ is a shared set of parameters between tasks, while $\tilde{\boldsymbol{\theta}}_t$ represents task-specific parameters introduced to "specialise" shared knowledge to the peculiarities of each task $t \in \{1, \ldots, T\}$. For instance, if a task parameter $\boldsymbol{\theta}_t \in \mathbb{R}^d$, our decomposition yields $\boldsymbol{\Theta}_{\text{sh}} \in \mathbb{R}^{d \times k}$, and $\tilde{\boldsymbol{\theta}}_t \in \mathbb{R}^{k \times 1}$ with $k$ representing the dimensions of the shared latent knowledge.

Solving the problem in Equation 2 amounts to determining both variational and model parameters, i.e., $\boldsymbol{\phi}_1, \ldots, \boldsymbol{\phi}_T$, and $\boldsymbol{\Theta}_{\text{sh}}$, and $\tilde{\boldsymbol{\theta}}_1, \ldots, \tilde{\boldsymbol{\theta}}_T$. We propose an expectation-maximisation style algorithm for computing each of the above free variables. Namely, in the E-step we solve for $\boldsymbol{\phi}_1, \ldots, \boldsymbol{\phi}_T$ while keeping $\boldsymbol{\Theta}_{\text{sh}}$, and $\tilde{\boldsymbol{\theta}}_1, \ldots, \tilde{\boldsymbol{\theta}}_T$ fixed. In the M-step, on the other hand, we determine $\boldsymbol{\Theta}_{\text{sh}}$ and $\tilde{\boldsymbol{\theta}}_1, \ldots, \tilde{\boldsymbol{\theta}}_T$ given the updated variational parameters. In both these steps, solving for the task-specific and variational parameters can be made efficient using parallelisation. Determining $\boldsymbol{\Theta}_{\text{sh}}$, however, requires knowledge of all tasks making it unscalable as the number of tasks grows large. To remedy this problem, we next propose a novel distributed Newton method with quadratic convergence guarantees[2]. Applying this method to determine $\boldsymbol{\Theta}_{\text{sh}}$ results in a highly scalable learner as shown in the following sections.

# 4 Scalable Multitask Reinforcement Learning

As mentioned earlier, the problem of determining $\boldsymbol{\Theta}_{\mathrm{sh}}$ can become computationally intensive with an increasing number of tasks. In this section, we devise a distributed Newton method for $\boldsymbol{\Theta}_{\mathrm{sh}}$ to aid in scalability. Given an updated variational (i.e., E-step) and fixed task-specific parameters, the optimisation problem for $\boldsymbol{\Theta}_{\mathrm{sh}}$ can be written as:

$$\max_{\boldsymbol{\Theta}_{\mathrm{sh}}} \sum_{t=1}^{T} \frac{1}{N_t} \left[ \sum_{i_t=1}^{N_t} \log \left[ p\left( \hat{\mathcal{R}}_t^{(i_t)} | \boldsymbol{\tau}_t^{(i_t)} \right) \times p_{\boldsymbol{\Theta}_{\mathrm{sh}}, \tilde{\boldsymbol{\theta}}_t} \left( \boldsymbol{\tau}_t^{(i_t)} \right) \right] \right] \equiv \max_{\boldsymbol{\Theta}_{\mathrm{sh}}} \sum_{t=1}^{T} \mathcal{J}_{\mathrm{MTRL}}^{(t)} \left( \boldsymbol{\Theta}_{\mathrm{sh}} \tilde{\boldsymbol{\theta}}_t \right), \quad (3)$$

where $i_t = \{1, \ldots, N_t\}$ denotes the index of trajectory $i$ for task $t \in \{1, \ldots, T\}$. The above equation omits functions independent of $\boldsymbol{\Theta}_{\mathrm{sh}}$, and estimates the variational expectation by sampling $N_t$ trajectories for each of the tasks.

Our scaling strategy is to allow for a distributed framework generalising to *any* topology of connected processors. Hence, we assume an undirected graph $\mathcal{G} = (\mathcal{V}, \mathcal{E})$ of computational units. Here, $\mathcal{V}$ denotes the set of nodes (processors) and $\mathcal{E}$ the set of edges. Similar to [11], we assume $n$ nodes connected via $|\mathcal{E}|$ edges. Contrary to their work, however, no specific node-ordering assumptions are imposed. Before writing the problem above in an equivalent distributed fashion, we firstly introduce "vec($\boldsymbol{A}$)" to denote the column-wise vectorisation of a matrix $\boldsymbol{A}$. This notation allows us to rewrite the by-product $\boldsymbol{\Theta}_{\mathrm{sh}} \tilde{\boldsymbol{\theta}}_t$ in terms of a vectorised version of the optimisation variable $\boldsymbol{\Theta}_{\mathrm{sh}}$, where $\mathrm{vec}(\boldsymbol{\Theta}_{\mathrm{sh}} \tilde{\boldsymbol{\theta}}_t) = (\tilde{\boldsymbol{\theta}}_t^\top \otimes \boldsymbol{I}_{d \times d}) \mathrm{vec}(\boldsymbol{\Theta}_{\mathrm{sh}}) \in \mathbb{R}^{d \times 1}$. Hence, the equivalent *distributed* formulation of Equation 3 is given by:

$$\min_{\boldsymbol{\Theta}_{\mathrm{sh}}^{(1)} : \boldsymbol{\Theta}_{\mathrm{sh}}^{(n)}} \sum_{i=1}^{n} \sum_{t=1}^{T_i} -\mathcal{J}_{\mathrm{MTRL}}^{(t)} \left( \left( \tilde{\boldsymbol{\theta}}_t^\top \otimes \boldsymbol{I}_{d \times d} \right) \mathrm{vec}(\boldsymbol{\Theta}_{\mathrm{sh}}^{(i)}) \right) \text{ s.t. } \boldsymbol{\Theta}_{\mathrm{sh}}^{(1)} = \cdots = \boldsymbol{\Theta}_{\mathrm{sh}}^{(n)}, \quad (4)$$

where $T_i$ is the total number of tasks assigned to node $i$ such that $\sum_{i=1}^{n} T_i = T$. Intuitively, the above is distributing Equation 3 among $n$ nodes, where each computes its local copy of $\boldsymbol{\Theta}_{\mathrm{sh}}$. For the distributed version to coincide with the centralised one, all nodes have to arrive to a consensus (in a fully distributed fashion) on the value of $\boldsymbol{\Theta}_{\mathrm{sh}}$. As such, a feasible solution for the distributed and centralised versions coincide making the two problems equivalent.

Now, we can apply any off-the-shelf distributed optimisation algorithm. Unfortunately, current techniques suffer from drawbacks prohibiting their direct usage for MTRL. Generally, there are two popular classes of algorithms for distributed optimisation. The first is sub-gradient based, while the second relies on a decomposition-coordination procedure. Sub-gradient algorithms proceed by taking a gradient step then followed by an averaging step at each iteration. The computation of each step is relatively cheap and can be implemented in a distributed fashion [22]. Though cheap to compute, the best known convergence rate of sub-gradient methods is slow given by $\mathcal{O}\left(1/\sqrt{K}\right)$ with $K$ being the total number of iterations [23, 24]. The second class of algorithms solve constrained problems by relying on dual methods. One of the well-known state-of-the-art methods from this class is the Alternating Direction Method of Multipliers (ADMM) [13]. ADMM decomposes the original problem to two subproblems which are then solved sequentially leading to updates of dual variables. In [23], the authors show that ADMM can be fully distributed over a network leading to improved convergence rates to the order of $\mathcal{O}\left(1/K\right)$. Recently, the authors in [11] applied the method [23] for distributed MTRL. In our experiments, we significantly outperform [11], especially in high-dimensional environments.

Much rate improvements can be gained from adopting second-order (Newton) methods. Though a variety of techniques have been proposed in [25–27], less progress has been made at leveraging ADMM's accuracy and convergence rate issues. In a recent attempt [25], the authors propose a distributed second-order method for general consensus by using the approach in [27] to compute the Newton direction. As detailed in Section 6, this method suffers from two problems. First, it fails to outperform ADMM and second, faces storage and computational deficiencies for large data sets, thus ADMM retains state-of-the-art status.

Next, we develop a distributed solver that outperforms others both theoretically and empirically. On the theory side, we develop the first distributed MTRL algorithm with provable *quadratic convergence guarantees*. On the empirical side, we demonstrate the superiority of our method on a variety of benchmarks.

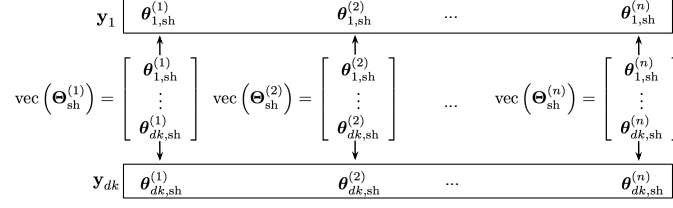

Figure 1: High-level depiction of our distribution framework for the shared parameters. Each of the vectors $\boldsymbol{y}_i$ holds the $i^{th}$ components of the shared parameters across all nodes $n$.

## 4.1 Laplacian-Based Distributed Multitask Reinforcement Learning

For maximum performance boost, we aim to have our algorithm exploit (locally) the structure of the computational graph connecting the processing units. To consider such an effect, we rewrite our distributed MTRL in terms of the graph Laplacian $\mathcal{L}$; a matrix that reflects the graph structure. Formally, the $\mathcal{L}$ is an $n \times n$ matrix such $\mathcal{L}(i,j) = \text{degree}(i)$ when $i = j$, -1 when $(i,j) \in \mathcal{E}$, and 0 otherwise. Of course, this matrix cannot be known to all the nodes in the network. We ensure full distribution by allowing each node to only access its local neighbourhood. To view the problem in Equation 4 from a graph topology perspective, we introduce a set of $dk$ vectors $\boldsymbol{y}_1, \ldots, \boldsymbol{y}_{dk}$, each in $\mathbb{R}^n$. The goal of these vectors is to hold the $i^{th}$ component of $\text{vec}(\boldsymbol{\Theta}_{\text{sh}}^{(1)}), \ldots, \text{vec}(\boldsymbol{\Theta}_{\text{sh}}^{(n)})$. This process is depicted in Figure 1, where, for instance, the first vector $y_1$ accumulates the first component of the shared parameters, $\boldsymbol{\theta}_{1,\text{sh}}^{(1)}, \ldots, \boldsymbol{\theta}_{1,\text{sh}}^{(n)}$, from all nodes.

We can now describe consensus on the copies of the shared parameters as consensus between the components of $\boldsymbol{y}_1, \ldots, \boldsymbol{y}_{dk}$. Clearly, the components in $\boldsymbol{y}_r$ coincide, if the $r^{th}$ component of the shared parameters equate across nodes. Hence, consensus between the components (for all $r$) corresponds to consensus on all dimensions of the shared parameters. This is exactly the constraint in Equation 4. One can think of a vector with equal components as that parallel to $\boldsymbol{1}$, namely, $\boldsymbol{y}_r = c_r \boldsymbol{1}$. Consequently, we can introduce the graph Laplacian in our constraints by having $\boldsymbol{y}_1, \ldots, \boldsymbol{y}_{dk}$ to be the solution of $\mathcal{L}\boldsymbol{y}_1 = \boldsymbol{0}, \ldots, \mathcal{L}\boldsymbol{y}_{dk} = \boldsymbol{0}$. This is true since the only solution to $\mathcal{L}\boldsymbol{v} = \boldsymbol{0}$ is a vector, $\boldsymbol{v}$, parallel to the vector of ones. Hence, a vector $\boldsymbol{y}_r$ satisfying the above system has to be of the form $c_r \boldsymbol{1}$, i.e., its components equate. Hence, we write:

$$\min_{\boldsymbol{y}_1 : \boldsymbol{y}_{dk}} \sum_{i=1}^{n} \sum_{t=1}^{T_i} -\mathcal{J}_{\text{MTRL}}^{(t)} \left( \left( \tilde{\boldsymbol{\theta}}_t^\top \otimes \boldsymbol{I}_{d \times d} \right) \tilde{\boldsymbol{y}}_i \right) \quad \text{s.t.} \quad \mathcal{L}\boldsymbol{y}_1 = \boldsymbol{0}, \ldots, \mathcal{L}\boldsymbol{y}_{dk} = \boldsymbol{0} \iff \boldsymbol{M}\boldsymbol{y} = 0, \quad (5)$$

with $\tilde{\boldsymbol{y}}_i = [\boldsymbol{y}_1(i), \ldots, \boldsymbol{y}_{dk}(i)]^\top$ denoting $\text{vec}(\boldsymbol{\Theta}_{\text{sh}}^{(i)})$, $\boldsymbol{M} = \boldsymbol{I}_{dk \times dk} \otimes \mathcal{L}$ a block-diagonal matrix of size $ndk \times ndk$ having Laplacian elements, and $\boldsymbol{y} \in \mathbb{R}^{ndk}$ a vector collecting $\boldsymbol{y}_1, \ldots, \boldsymbol{y}_{dk}$.

## 4.2 Solution Methodology

The problem in Equation 5 is a constrained optimisation one that can be solved by descending (in a distributed fashion) in the dual function. Though adopting second-order techniques (e.g., Newton iteration) can lead to improved convergence speeds, direct application of standard Newton is difficult as we require a *distributed procedure* to *accurately* compute the direction of descent[3].

In the following, we propose an accurate and scalable distributed Newton method. Our solution is decomposed in two steps. First, we write the constraint problem as an unconstraint one by introducing the dual functional to Equation 5. Second, we exploit the symmetric diagonally dominant (SDD) property of the Hessian, previously proved for a broader setting in Lemma 2 of [28], by developing a Chebyshev solver to compute the Newton direction. To formulate the dual, we introduce a vector of Lagrange multipliers $\boldsymbol{\lambda} = [\boldsymbol{\lambda}_1^\top, \ldots, \boldsymbol{\lambda}_{dk}^\top]^\top \in \mathbb{R}^{ndk}$, where $\boldsymbol{\lambda}_i \in \mathbb{R}^n$ is a vector of multipliers, one for each dimension of $\text{vec}(\boldsymbol{\Theta}_{\text{sh}})$. For fully distributed computations, we assume each node to only store its corresponding components $\boldsymbol{\lambda}_1(i), \ldots, \boldsymbol{\lambda}_{dk}(i)$. After deriving the Lagrangian, we can write

the dual function $q(\boldsymbol{\lambda})$ as [4]:

$$q(\boldsymbol{\lambda}) = \sum_{i=1}^{n} \inf_{\boldsymbol{y}_1(i):\boldsymbol{y}_{dk}(i)} \left( \sum_{t=1}^{T_i} -\mathcal{J}_{\mathrm{MTRL}}^{(t)} \left( \left( \tilde{\boldsymbol{\theta}}_t^{\top} \otimes \boldsymbol{I}_{d \times d} \right) \tilde{\boldsymbol{y}}_i \right) + \boldsymbol{y}_1(i)[\mathcal{L}\boldsymbol{\lambda}_1]_i + \cdots + \boldsymbol{y}_{dk}(i)[\mathcal{L}\boldsymbol{\lambda}_{dk}]_i \right),$$

which is clearly separable across the computational nodes in $\mathcal{G}$. Before discussing the SDD properties of the dual Hessian, we still require a procedure that allows us to infer about the primal (i.e., $\boldsymbol{y}$) given updated parameters $\boldsymbol{\lambda}$. We recognise that primal variables can be found as the solution to the following system of equations:

$$\frac{\partial f_i(\cdot)}{\partial \boldsymbol{y}_1(i)} = -[\mathcal{L}\boldsymbol{\lambda}_1]_i, \quad \ldots \quad , \frac{\partial f_i(\cdot)}{\partial \boldsymbol{y}_{dk}(i)} = -[\mathcal{L}\boldsymbol{\lambda}_{dk}]_i \text{ where } f_i(\cdot) = \sum_{t=1}^{T_i} -\mathcal{J}_{\mathrm{MTRL}}^{(t)} \left( \left( \tilde{\boldsymbol{\theta}}_t^{\top} \otimes \boldsymbol{I}_{d \times d} \right) \tilde{\boldsymbol{y}}_i \right).$$

(6)

It is also clear that Equation 6 is locally defined for every node $i \in \mathcal{V}$ since for each $r = 1, \ldots, dk$, we have: $-[\mathcal{L}\boldsymbol{\lambda}_r]_i = \sum_{j \in \mathcal{N}(i)} \lambda_r(j) - d(i)\lambda_r(i)$, where $\mathcal{N}(i)$ is the neighbourhood of node $i$. As such, each node $i$ can construct its own system of equations by collecting $\{\lambda_1(j), \ldots, \lambda_{dk}(j)\}$ from its neighbours *without the need for full communication*. These can then be locally solved for determining the primal variables[5].

As mentioned earlier, we update $\boldsymbol{\lambda}$ using a distributed Newton method. At every iteration $s$ in the optimisation algorithm, the descent direction is thus computed to be the solution of $\boldsymbol{H}(\boldsymbol{\lambda}_s)\boldsymbol{d}_s = -\boldsymbol{g}_s$, where $\boldsymbol{H}(\boldsymbol{\lambda}_s)$ is the Hessian, $\boldsymbol{d}_s$ the Newton direction, and $\boldsymbol{g}_s$ the gradient. The Hessian and the gradient of our objective are given by:

$$\boldsymbol{H}(\boldsymbol{\lambda}_s) = -\boldsymbol{M} \left[ \sum_{i=1}^{n} \sum_{t=1}^{T_i} -\nabla^2 \mathcal{J}_{\mathrm{MTRL}}^{(t)} \left( \boldsymbol{y}(\boldsymbol{\lambda}_s) \right) \right]^{-1} \boldsymbol{M} \quad \text{and} \quad \nabla q(\boldsymbol{\lambda}_s) = \boldsymbol{M}\boldsymbol{y}(\boldsymbol{\lambda}_s).$$

Unfortunately, inverting $\boldsymbol{H}(\boldsymbol{\lambda}_s)$ to determine the Newton direction is not possible in a distributed setting since computing the inverse requires global information. Given the form of $\boldsymbol{M}$ and following the results in [28], one can show that the above Hessian exhibits the SDD property. Luckily, this property can be exploited for a distributed solver as we show next.

The story of computing an approximation to the exact solution of an SDD system of linear equation starts with standard splitting of symmetric matrices. Given a symmetric matrix[6] $\boldsymbol{H}$ the standard splitting is given by $\boldsymbol{H} = \boldsymbol{D}_0 - \boldsymbol{A}_0$, where $\boldsymbol{D}_0$ is a diagonal matrix that consists of diagonal elements in $\boldsymbol{H}$, while $\boldsymbol{A}_0$ is a matrix collecting the negate of the off-diagonal components in $\boldsymbol{H}$. As the goal is to determine a solution of the SDD system, we will be interested in inverses of $\boldsymbol{H}$. Generalising the work in [29], we recognise that the inverse can be written as:

$$(\boldsymbol{D}_0 - \boldsymbol{A}_0)^{-1} \approx \boldsymbol{D}_0^{-\frac{1}{2}} \prod_{\ell=0}^{\mathcal{O}(\log m)} \left[ \boldsymbol{I} + \left[ \boldsymbol{D}_0^{-\frac{1}{2}} \boldsymbol{A}_0 \boldsymbol{D}_0^{-\frac{1}{2}} \right]^{2^{\ell}} \right] \boldsymbol{D}_0^{-\frac{1}{2}} = \hat{\mathcal{P}}_m(\boldsymbol{H}),$$

where $\hat{\mathcal{P}}_m(\boldsymbol{H})$ is a polynomial of degree $m \sim \kappa(\boldsymbol{H})$ of matrix $\boldsymbol{H}$. All computations do not need access to the Hessian nor its inverse. We can describe these only using local Hessian vector products, hence allowing for fast implementation using automatic differentiation. Hence, the goal of the Newton update is to find a solution of the form $\boldsymbol{d}_s^{(m)} = \mathcal{P}_m(\boldsymbol{H}(\boldsymbol{\lambda}_s))\nabla q(\boldsymbol{\lambda}_s)$ such that $\boldsymbol{d}_s^{(m)}$ is an $\epsilon$-close solution to $\boldsymbol{d}_s^{\star}$. Consequently, the differential $\boldsymbol{d}_s^{(m)} - \boldsymbol{d}_s^{\star}$ can be written as:

$$\boldsymbol{d}_s^{(m)} - \boldsymbol{d}_s^{\star} = [\boldsymbol{H}(\boldsymbol{\lambda}_s)\mathcal{P}_m(\boldsymbol{H}(\boldsymbol{\lambda}_s)) - \boldsymbol{I}] \boldsymbol{d}_s^{\star} = -\mathcal{Q}_m(\boldsymbol{H}(\boldsymbol{\lambda}_s))\boldsymbol{d}_s^{\star},$$

where $\mathcal{Q}_m(\boldsymbol{H}(\boldsymbol{\lambda}_s)) = -\boldsymbol{H}(\boldsymbol{\lambda}_s)\mathcal{P}_m(\boldsymbol{H}(\boldsymbol{\lambda}_s)) + \boldsymbol{I}$. Therefore, instead of seeking $\mathcal{P}_m(\cdot)$, one can think of constructing polynomials $\mathcal{Q}_m(\cdot)$ that reduce the term $\boldsymbol{d}_s^{(m)} - \boldsymbol{d}_s^{\star}$ as fast as possible. This can be formalised in terms of the properties of $\mathcal{Q}_m(\cdot)$ by requiring the polynomial to have a minimal degree, as well as satisfying the following for a precision parameter $\epsilon$: $\mathcal{Q}_m(0) = 1$ and $|\mathcal{Q}_m(\mu_i)| \leq \epsilon$,

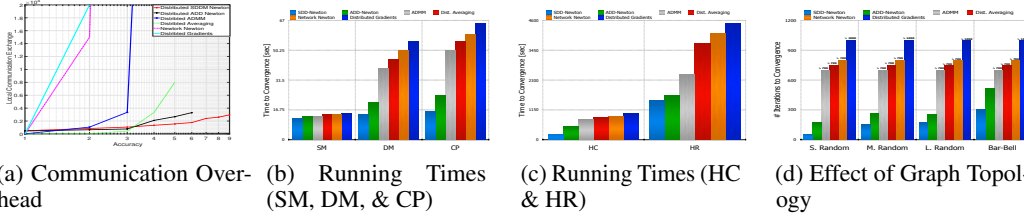

| (a) Communication Over-head | (b) Running Times (SM, DM, & CP) | (c) Running Times (HC & HR) | (d) Effect of Graph Topology |
|---|---|---|---|

Figure 2: (a) Communication overhead in the HC case. Our method has an increase proportional to the condition number of the graph, which is slower compared to the other techniques. (b) and (c) Running times till convergence to a threshold of $10^{-5}$. (d) Number of iterations for a $10^{-5}$ consensus error on the HC dynamical system on different graph topologies.

with $\mu_i$ being the $i^{th}$ smallest eigenvalue of $\boldsymbol{H}(\boldsymbol{\lambda}_s)$. The first condition is a result of observing $\mathcal{Q}_m(z) = -z\mathcal{P}_m(z) + 1$, while the second guarantees an $\epsilon$-approximate solution:

$$||\boldsymbol{d}_k^{(m)} - \boldsymbol{d}_k^\star||_{\boldsymbol{H}(\boldsymbol{\lambda}_s)}^2 \leq \max_i |\mathcal{Q}_m(\mu_i)|^2 ||\boldsymbol{d}_s^\star||_{\boldsymbol{H}(\boldsymbol{\lambda}_s)}^2 \leq \epsilon^2 ||\boldsymbol{d}_s^\star||_{\boldsymbol{H}(\boldsymbol{\lambda}_s)}^2.$$

In other words, finding $\mathcal{Q}_m(z)$ that has minimal degree and satisfies the above two conditions guarantees an efficient and $\epsilon$-close solution to $\boldsymbol{d}_s^\star$. Chebyshev polynomials of the first kind satisfy our requirements. Their form is defined as $T_m(z) = \cos(m\arccos(z))$ if $z \in [-1, 1]$, and $\frac{1}{2}((z + \sqrt{z^2 - 1})^m + (z - \sqrt{z^2 - 1})^m)$ otherwise. Interestingly, $|T_m(z)| \leq 1$ on $[-1, 1]$, and among all polynomials of degree $m$ with a leading coefficient 1, the polynomial $\frac{1}{2^{m-1}}T_m(z)$ acquires its minimal and maximal values on this interval (i.e., sharpest increase outside the range $[-1, 1]$). We posit that a good candidate is $\mathcal{Q}_m^\star(z) = T_m\left(\frac{\mu_N + \mu_2 - 2z}{\mu_N - \mu_2}\right) / T_m\left(\frac{\mu_N + \mu_2}{\mu_N - \mu_2}\right)$, with $\mu_i$ being the $i^{th}$ smallest eigenvalue of symmetric matrix $\boldsymbol{H}$ describing the system of linear equations. First, it is easy to see that when $z = 0$, these polynomials attain a value of unity (i.e., $\mathcal{Q}_m^\star(0) = 1$). Secondly, it can be shown that for any $s$ and $z \in [\mu_2, \mu_{ndk}]$, $|\mathcal{Q}^\star(z)|^2$ is bounded as $|\mathcal{Q}^\star(z)|^2 \leq 4\exp\left(-4m/\sqrt{\kappa(\boldsymbol{H})} + 1\right)$.

Therefore, choosing the approximate solution as[7] $\boldsymbol{d}_s^{(m)} = -\boldsymbol{H}(\boldsymbol{\lambda}_s)^{-1}\left(\boldsymbol{I} - \mathcal{Q}_m(\boldsymbol{H}(\boldsymbol{\lambda}_s))\right)\boldsymbol{g}_s$ guarantees an $\epsilon$-close solution. Please note that by exploiting the recursive properties of Chebyshev polynomials we can derive an approximate solution without the need to compute $\boldsymbol{H}^{-1}$ explicitly. In addition to the time and message complexities of this new solver, other implementation details can be found in the appendix. We now show quadratic convergence of the distributed Newton method:

**Theorem 1.** *Distributed Newton method using the Chebyshev solver exhibits the following two convergence phases for some constants $c_1$ and $c_2$:*

**Strict Decrease:** *if $||\nabla q(\boldsymbol{\lambda}_s)||_2 > c_1$, then $||\nabla q(\boldsymbol{\lambda}_s)||_2 - ||\nabla q(\boldsymbol{\lambda}_s)||_2 \leq c_2 \frac{\mu_2^4(\mathcal{L})}{\mu_n^3(\mathcal{L})}$*

**Quadratic Decrease:** *if $||\nabla q(\boldsymbol{\lambda}_s)||_2 \leq c_1$, then for any $l \geq 1$: $||\nabla q(\boldsymbol{\lambda}_{s+l})||_2 \leq \frac{2c_1}{2^{2^l}} + \mathcal{O}(\epsilon)$*

## 5 Experiments & Results

We conducted two sets of experiments to compare against distributed and multitask learning methods. On the distributed side, we evaluated our algorithm against five other approaches being: 1) ADD [27], 2) ADMM [11], 3) distributed averaging [30], 4) network-newton [26, 25], and 5) sub-gradients. We are chiefly interested in the convergence speeds of both the objective value and consensus error, as well as the communication overhead and running times of these approaches. The comparison against [11] (which we title as distributed ADMM in the figures) allows us to understand whether we surpass state-of-the-art, while that against ADD and network-newton sheds the light on the accuracy of our Newton's direction approximation. When it comes to online methods, we compare our performance in terms of jump-start and asymptotic performance to policy gradients [31, 32], PG-ELLA [10], and GO-MTL [33]. Our experiments ran on five systems, simple mass (SM), double mass (DM), cart-pole (CP), helicopter (HC), and humanoid robots (HR).

We followed the experimental protocol in [10, 33] where we generated 5000 SM, 500 DM, and 1000 CP tasks by varying the dynamical parameters of each of the above systems. These tasks were then

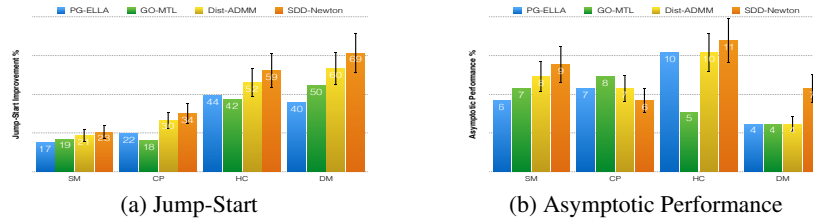

| (a) Jump-Start | (b) Asymptotic Performance |

Figure 3: Demonstration of jump-start and asymptotic results.

distributed over graphs with edges generated uniformly at random. Namely, a graph of 10 nodes and 25 edges was used for both SM and DM experiments, while a one with 50 nodes and 150 edges for the CP. To distribute our computations, we made use of MATLAB's parallel pool running on 10 nodes. For all methods, tasks were assigned evenly across 10 agents[8]. An $\epsilon = 1/100$ was provided to the Chebyshev solver for determining the approximate Newton direction in all cases. Step-sizes were determined separately for each algorithm using a grid-search-like technique over $\{0.01, \ldots, 1\}$ to ensure best operating conditions. Results reporting improvements in the consensus error (i.e., the error measuring the deviation from agreement among the nodes) can be found in the appendix due to space constraints.

**Communication Overhead & Running Times:** It can be argued that our improved results arrive at a high communication cost between processors. This may be true as our method relies on an SDD-solver while others allow for only few messages per iteration. We conducted an experiment measuring local communication exchange with respect to accuracy requirements. Results on the HC system, reported in Figure 2a, demonstrate that this increase is negligible compared to other methods. Clearly, as accuracy demands increase so does the communication overhead of all algorithms. Distributed SDD-Newton has a growth rate proportional to the condition number of the graph being much slower compared to the exponential growth observed by other techniques. Having shown small increase in communication cost, now we turn our attention to assess running times to convergence on all dynamical systems. Figures 2b and 2c report running times to convergence computed according to a $10^{-5}$ error threshold. All these experiments were run on a small random graph of 20 nodes and 50 edges. Clearly, our method is faster when compared with others in both cases of low and high-dimensional policies. A final question to be answered is the effect of different graph topologies on the performance of SDD-Newton. Taking the HC benchmark, we generated four graph topologies representing small (S. Random), medium (M. Random), and large (L. Random) random networks, and a bar-bell graph with nodes varying from 10 to 150 and edges from 25 to 250. The bar-bell contained 2 cliques formed by 10 nodes each and a 10 node line graph connecting them. We then measured the number of iterations required by all algorithms to achieve a consensus error of $10^{-5}$. Figure 2d reports these results showing that our method is again faster than others.

**Benchmarking Against RL:** We finally assessed our method in comparison to current MTRL literature, including PG-ELLA [10] and GO-MTL [33]. For the experimental procedure, we followed the technique described in [11], where the reward function was given by $-\sqrt{x_h - x_{\text{ref}}}$, with $x_{\text{ref}}$ being the reference state. As base-learners we used policy gradients as detailed in [34], which acquired 1000 trajectories with a length of 150 each. We report jump-start and asymptotic performance in Figures 3a and 3b. These results show that our method can outperform others in terms of jump-start and asymptotic performance while requiring fewer iterations. Moreover, it is clear that our method outperforms streaming models, e.g., PG-ELLA.

# 6   Conclusions & Future Work

We proposed a distributed solver for multitask reinforcement learning with quadratic convergence. Our next steps include developing an incremental version of our algorithm using generalised Hessians, and conducting experiments running on true distributed architectures to quantify the trade-off between communication and computation.

## Footnotes

[1] Please note that we leave exploring other forms of the variational distribution as an interesting direction for future work.

[2] Contrary to stochastic variational inference, we are not restricted to exponential family distributions

[3]It is worth noting that some techniques for determining the Newton direction in a distributed fashion exist. These techniques, however, are inaccurate, see Section 5.

[4]Please notice that for a dual function we use notation $q(\boldsymbol{\lambda})$ and for the variational distribution $q_{\phi_t}(\tau_t)$

[5]Please note that for the case of log-concave policies, we can determine the relation between primal and dual variables in closed form by simple algebraic manipulation.

[6]Please note that we use $\boldsymbol{H}$ to denote $\boldsymbol{H}(\boldsymbol{\lambda}_s)$.

[7]Please note that the solution of this system can be split into $dk$ linear systems that can be solved efficiently using the distributed Chebyshev solver. Due to space constraints these details can be found in the appendix.

[8]When graphs grew larger, nodes were grouped together and provided to one processor.

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
