[Supplementary Material]

# A  Algorithmic Details

To summarise the main steps of the submission, we now detail a pseudo-code for distributed multitask reinforcement learning.

---

**Algorithm 1** Distributed Multitask Reinforcement Learning

---

**Input:** A set of $T$ MDPs, dimensions of the latent space $k$, parameter initialisation for $\boldsymbol{\Theta}_{\text{sh}}$, $\tilde{\boldsymbol{\theta}}_1, \ldots, \tilde{\boldsymbol{\theta}}_T, \boldsymbol{\phi}_1, \ldots, \boldsymbol{\phi}_T$, precision parameter $\epsilon$.

**Step 1:** Optimise for the variational parameters $\boldsymbol{\phi}_1, \ldots, \boldsymbol{\phi}_T$ by solving:

$$\max_{\boldsymbol{\phi}_1:\boldsymbol{\phi}_T} \sum_{t=1}^{T} \mathbb{E}_{q_{\boldsymbol{\phi}_t}(\boldsymbol{\tau}_t)} \left[ \log p \left( \hat{\mathcal{R}}_t | \boldsymbol{\tau}_t \right) \right] - \sum_{t=1}^{T} \mathcal{D}_{\text{KL}} \left( q_{\boldsymbol{\phi}_t}(\boldsymbol{\tau}_t) || p_{\boldsymbol{\theta}_t}(\boldsymbol{\tau}_t) \right).$$

**Step 2:** Given updated $\boldsymbol{\phi}_1, \ldots, \boldsymbol{\phi}_T$, solve for $\boldsymbol{\Theta}_{\text{sh}}$ as follows:
   **Step 2.1:** Distribute tasks among a graph $\mathcal{G}$ of $n$ computational units
   **Step 2.2:** Update $\boldsymbol{\Theta}_{\text{sh}}$ using our distributed Newton method up-to precision $\epsilon$ (Section 4.2)
**Step 3:** Given updated $\boldsymbol{\phi}_1, \ldots, \boldsymbol{\phi}_T$, and $\boldsymbol{\Theta}_{\text{sh}}$, determine the task specific coefficients by solving:

$$\max_{\boldsymbol{\theta}_1:\boldsymbol{\theta}_T} \sum_{t=1}^{T} \mathbb{E}_{q_{\boldsymbol{\phi}_t}(\boldsymbol{\tau}_t)} \left[ \log p \left( \hat{\mathcal{R}}_t | \boldsymbol{\tau}_t \right) \right] - \sum_{t=1}^{T} \mathcal{D}_{\text{KL}} \left( q_{\boldsymbol{\phi}_t}(\boldsymbol{\tau}_t) || p_{\boldsymbol{\theta}_t}(\boldsymbol{\tau}_t) \right).$$

**Output:** Variational, shared, and task specific parameters.

---

# B  Additional Experiments

This section details additional experimental results reflecting consensus errors on various systems. We ran our experiments on the benchmarks depicted in Figure 1.

## B.1  Additional Results

Clearly, our algorithm converges faster than others to low-consensus error and optimal objective values.

# C  Theoretical Guarantees

For the clarity of the presentation we split the Appendix in several sections. In the first section, we provide theoretical analysis for Distributed Chebyshev Solver for solving SDD systems.In the second section we provide the convergence analysis for Distributed Newton network and prove Theorem **??**

Figure 1: A high-level depiction of the benchmark dynamical systems used in our experiments.

| | SDD-Newton | ADD-Newton | Netw-Newton | Dist-ADMM | Dist-Average | Dist-Gradient |
|---|---|---|---|---|---|---|
| **SM** | $10^1$ | $10^2$ | $\sim 10^2$ | $\sim 10^4$ | $\sim 10^4$ | $\sim 10^5$ |
| **DM** | $10^2$ | $10^3$ | $\sim 10^5$ | $\sim 10^4$ | $\sim 10^5$ | $\sim 10^5$ |
| **CP** | $10^3$ | $\sim 10^4$ | $\sim 10^5$ | $\sim 10^5$ | $\sim 10^5$ | $\sim 10^5$ |

Figure 2: Number of iterations needed for convergence to low consensus showing that our method outperforms state-of-the-art techniques.

(a) Obj. HC

(b) Con. HC

(c) Obj. HR

(d) Con. HR

Figure 3: Figures (a) and (b) report the objective value and consensus error versus iterations on HC systems. Figures (c) and (d) demonstrate the same criteria on the humanoid tasks. In all these cases, our method outperforms others in literature.

## C.1 Distributed Chebyshev Solver

The approximated solution vector $\boldsymbol{d}_s^{(m)}$ can be represented as a concatenation of $dk$ vectors: $\boldsymbol{d}_s^{(m)} = \boldsymbol{d}_s^{(m),1,\mathsf{T}}, \ldots, \boldsymbol{d}_s^{(m),dk,\mathsf{T}}]^{\mathsf{T}}$, where each vector $\boldsymbol{d}_s^{(m),i}$ is an $\epsilon-$approximated solution of the system

$$\mathcal{L}\boldsymbol{d}_s^{(m),i} = \boldsymbol{b}_s^i \tag{1}$$

with vector $\boldsymbol{b}_s^i$ is the $i^{th}$ chunk of vector $\boldsymbol{b}_s = \sum_{i=1}^n \sum_{t=1}^{T_i} \nabla^2 \mathcal{J}_{MTRL}^{(t)}(\boldsymbol{y}(\boldsymbol{\lambda_s}))\boldsymbol{y}(\boldsymbol{\lambda})_s$. Please notice, that each vector $\boldsymbol{b}_s^i \in \mathbb{R}^n$ and it is distributed across the nodes of graph $\mathcal{G}$. Indeed, the $r^{th}$ component of this vector can be computed as follows:

$$[\boldsymbol{b}_s^i]_r = \sum_{j=1}^{dk} \frac{\partial^2 \left[ \sum_{t=1}^{T_i} \mathcal{J}_{MTRL}^{(t)}(\boldsymbol{y}(\boldsymbol{\lambda_s})) \right]}{\partial [\boldsymbol{y}_i]_r \partial [\boldsymbol{y}_j]_r} [\boldsymbol{y}(\boldsymbol{\lambda_s})_j]_r \tag{2}$$

where we represented primal variable $\boldsymbol{y}(\boldsymbol{\lambda_s})$ as concatenation $\boldsymbol{y}(\boldsymbol{\lambda_s}) = \left[\boldsymbol{y}(\boldsymbol{\lambda_s})_1^{\mathsf{T}}, \ldots, \boldsymbol{y}(\boldsymbol{\lambda_s})_{dk}^{\mathsf{T}}\right]^{\mathsf{T}}$. Indeed, Equation (2) can be computed locally by node $r \in \mathcal{V}$ because it stores the $r^{th}$ components of vectors $\boldsymbol{y}(\boldsymbol{\lambda_s})_1, \ldots, \boldsymbol{y}(\boldsymbol{\lambda_s})_{dk}$ as well as it stores the local primal objective $\sum_{t=1}^{T_i} \mathcal{J}_{MTRL}^{(t)}(\boldsymbol{y}(\boldsymbol{\lambda_s}))$.

This observation allows us to distribute the computation of an $\epsilon-$approximated solution vector $\boldsymbol{d}_s^{(m),i}$ using Chebyshev polynomials:

$$\boldsymbol{d}_s^{(m),i} = \mathcal{L}^{\dagger}(\boldsymbol{I} - \mathcal{Q}_m(\mathcal{L}))\boldsymbol{b}_k^i, \qquad i = 1\ldots, dk. \tag{3}$$

where $m = \lceil \frac{1}{2}\left(\sqrt{\kappa(\mathcal{L})} + 1\right) \ln \frac{2}{\epsilon} \rceil$ and

$$\mathcal{Q}_m(z) = \frac{T_m\left(\frac{(\mu_n + \mu_1) - 2z}{\mu_n - \mu_1}\right)}{T_m\left(\frac{\mu_n + \mu_1}{\mu_n - \mu_1}\right)} \tag{4}$$

where $\mu_1, \mu_n$ are smallest and largest non-zero eigenvalues of graph Laplacian $\mathcal{L}$. Please notice , that for any $z \in [\mu_1, \mu_p]$ we have

$$|\mathcal{Q}_m(z)|^2 \leq T_m^{-2}\left(\frac{\mu_p + \mu_1}{\mu_p - \mu_1}\right) = T_m^{-2}\left(\frac{\kappa(\boldsymbol{L}_{\mathcal{G}}) + 1}{\kappa(\mathcal{L}) - 1}\right) \leq 4e^{-\frac{4m}{\sqrt{\kappa(\mathcal{L})} + 1}}$$

where $\kappa(\mathcal{L}) = \frac{\mu_p}{\mu_1}$. Let $\boldsymbol{d}_s^{*,i}$ be the exact solution of system (1), then a solution vector in Equation (3) satisfies $||\boldsymbol{d}_s^{(m),i} - \boldsymbol{d}_s^{*,i}||_{\mathcal{L}}^2 \leq 4e^{-\frac{4m}{\sqrt{\kappa(\mathcal{L})} + 1}}||\boldsymbol{d}_s^{*,i}||_{\mathcal{L}}^2$. Hence, by choosing degree $m = \lceil \frac{1}{2}(\sqrt{\kappa(\mathcal{L})} + 1) \ln \frac{2}{\epsilon} \rceil$ the solution vector

$$\boldsymbol{d}_s^{(m),i} = \mathcal{L}^{\dagger}\left[\frac{T_m\left(\frac{\mu_p + \mu_1}{\mu_p - \mu_1}\right)\boldsymbol{I} - T_m\left(\frac{((\mu_p + \mu_1)\boldsymbol{I} - 2\mathcal{L})}{\mu_p - \mu_1}\right)}{T_m\left(\frac{\mu_p + \mu_1}{\mu_p - \mu_1}\right)}\right]\boldsymbol{b}_s^i, \qquad i = 1, \ldots, dk \tag{5}$$

satisfies the $\epsilon-$accuracy requirement: $||\boldsymbol{d}_s^{(m),i} - \boldsymbol{d}_s^{*,i}||_{\mathcal{L}}^2 \leq \epsilon||\boldsymbol{d}_s^{*,i}||_{\mathcal{L}}$.

Having proposed an approximate solution $\boldsymbol{d}_s^{(m),i}$, at this stage we are ready to commence with the distributed implementation of our solver. However, we recognize the following two challenges hindering its direct distributed implementation. First, we note that computing the minimum and maximum non-zero eigenvalues of $\mathcal{L}$ requires global information. The second relates to the product with $\mathcal{L}^{\dagger}$ needed in Equation 5. Here we detail the solutions to above two problems for the case when $\mathcal{L}$ is graph Laplacian and derive our distributed SDD solver, which is used later to compute the Newton direction

1. **Parameters $\mu_1$ and $\mu_p$.** As clear from the previous section, our method requires the computation of the second-minimum and maximum eigenvalues of $\mathcal{L}$. The computation of these, however, requires global information and hence are difficult to determine in a distributed fashion. As a substitute for the exact values of $\mu_1$ and $\mu_p$, one can use the well-known eigenvalue bounds determined as

$$\mu_1 \geq \underline{\mu} = \frac{4}{n^2}$$
$$\mu_p \leq \bar{\mu} = 2n$$

2. **Multiplication on $\mathcal{L}^{\dagger}$.** We start by noting that the second issue faced relates to the computational inefficiency when attempting to compute the coefficients of $T_m\left(\frac{\mu_p + \mu_1}{\mu_p - \mu_1}\right)\boldsymbol{I} - T_m\left(\frac{((\mu_p + \mu_1)\boldsymbol{I} - 2\mathcal{L})}{\mu_p - \mu_1}\right)$ where performing it naively will potentially lead to linear dependency on the condition number of the processing graph. To illustrate, let us, in fact, consider the naive approach by assuming that each node $i$ has access to the following decomposition of $T_m(z)$:

$$T_m(z) = 1 + \alpha_1 z + \alpha_2 z^2 + \ldots + \alpha_m z^m$$

where $\alpha_1, \ldots, \alpha_m$ are coefficients one for each power of the polynomial. For ease of exposition, let us further denote

$$c_1 = \frac{\bar{\mu} + \underline{\mu}}{\bar{\mu} - \underline{\mu}} \qquad c_2 = \frac{2}{\bar{\mu} - \underline{\mu}}$$

Using the above, the numerator in Equation (5): can be written as

$$\mathcal{L}^\dagger \left[ T_m(c_1)\boldsymbol{I} - T_m(c_1\boldsymbol{I} - c_2\mathcal{L}) \right] \boldsymbol{b}_s^i =$$

$$\mathcal{L}^\dagger \left[ \sum_{\nu=1}^{m} \alpha_\nu c_1^\nu \boldsymbol{I} - \sum_{\nu=1}^{m} \alpha_\nu (c_1\boldsymbol{I} - c_2\mathcal{L})^\nu \right] \boldsymbol{b}_s^i =$$

$$\mathcal{L}^\dagger \left[ \sum_{\nu=1}^{m} \alpha_\nu \left[ (c_1\boldsymbol{I})^\nu - (c_1\boldsymbol{I} - c_2\mathcal{L})^\nu \right] \right] \boldsymbol{b}_s^i$$

The term $c_1^\nu \boldsymbol{I})$ is easy to compute. The second, on the other hand, can be computed by rewriting the term $(c_1\boldsymbol{I} - c_2\mathcal{L})^i$ explicitly in terms of $\mathcal{L}$ for each node $i$. Unfortunately, this procedure is inefficient as it boils-down to a total of $\mathcal{O}(m^2)$ matrix vector multiplications of the form $\mathcal{L}\boldsymbol{u}$. Taking into account the expression for $m$, we end up with an algorithm exhibiting linear dependency on the condition number $\kappa(\mathcal{L})$. Instead, our goal is to show that solution in (5) can be computed in fully distributed way in $\mathcal{O}(m)$ rounds. The crucial property for us here is the recursive relation of Chebyshev polynomials:

$$T_0(z) = 1, \qquad\qquad\qquad\qquad (6)$$
$$T_1(z) = z,$$
$$T_\ell(z) = 2zT_{\ell-1}(z) - T_{\ell-2}(z)$$

Denote by

$$\boldsymbol{\Delta}_\ell = \mathcal{L}^\dagger \left[ T_\ell(c_1)\boldsymbol{I} - T_\ell(c_1\boldsymbol{I} - c_2\mathcal{L}) \right] \boldsymbol{b}_s^i$$
$$\boldsymbol{\Omega}_\ell = T_\ell(c_1\boldsymbol{I} - c_2\mathcal{L})\boldsymbol{b}_s^i$$
$$\Theta_\ell = T_\ell(c_1)$$

Therefore, the solution vector (5) can be written as $\boldsymbol{d}_s^{(m),i} = \frac{\boldsymbol{\Delta}_m}{\Theta_m}$ and recursive relation gives:

$$\boldsymbol{\Delta}_\ell = 2c_1\boldsymbol{\Delta}_{\ell-1} - \boldsymbol{\Delta}_{\ell-2} + 2c_2\boldsymbol{\Omega}_{\ell-1} \qquad\qquad (7)$$
$$\boldsymbol{\Omega}_\ell = 2(c_1\boldsymbol{I} - c_2\mathcal{L})\boldsymbol{\Omega}_{\ell-1} - \boldsymbol{\Omega}_{\ell-2}$$
$$\Theta_\ell = 2c_1\Theta_{\ell-1} - \Theta_{\ell-2}$$

with initials given by:

$$\boldsymbol{\Delta}_1 = c_2\boldsymbol{b}_s^i \qquad\qquad \boldsymbol{\Omega}_1 = [c_1\boldsymbol{I} - c_2\mathcal{L}]\boldsymbol{b}_s^i \qquad\qquad \Theta_1 = c_1$$
$$\boldsymbol{\Delta}_0 = \boldsymbol{0} \qquad\qquad \boldsymbol{\Omega}_0 = \boldsymbol{b}_k^i \qquad\qquad \Theta_0 = 1$$

Algorithm 2 summarizes these results and provides a fully distributed computation of vector (5) in $\mathcal{O}(m)$ rounds. Clearly, lines 8-10 are executing relations (7) in a fully distributed way. Indeed, each matrix vector multiplication $(c_i\boldsymbol{I} - c_2\mathcal{L})\boldsymbol{u}$ can be computed locally by a single message exchange between the neighboring nodes. Moreover, the total number of such multiplications is bounded by $\mathcal{O}(m)$ and this fact establishes the following

**Theorem 1.** *The distributed SDD solver described in Algorithm 2 uses local communication exchange to compute an $\epsilon$-approximate solution of the SDD system (1) in the following number of rounds*

$$\mathcal{O}\left( \sqrt{\kappa(\mathcal{L})} \log\left(\frac{1}{\epsilon}\right) \right) \leq \mathcal{O}\left( \sqrt{nd_{max}diam(\mathcal{G})} \log\left(\frac{1}{\epsilon}\right) \right)$$

*where $\kappa(\mathcal{L})$ is condition number of $\mathcal{G}$, $d_{max}, diam(\mathcal{G})$ are its maximal degree and diameter.*

This result provides us with time complexity $\mathcal{O}\left( d_{max}\sqrt{\kappa(\mathcal{L})} \log\left(\frac{1}{\epsilon}\right) \right)$ and message complexity $\mathcal{O}\left( |\mathcal{E}|\sqrt{\kappa(\mathcal{L})} \log\left(\frac{1}{\epsilon}\right) \right)$.

---

**Algorithm 2** : **Chebyshev SDD Solver**

---

1: **Input:** The $r^{th}$ row of graph Laplacian $\mathcal{L}$, the $r^{th}$ component of vector $\boldsymbol{b}_k^i$, precision parameter $\epsilon$.
2: **Output:** The $r^{th}$ components of $\epsilon-$ approximate solution $\boldsymbol{d}_s^{(m),i}$.
3: Set $\bar{\mu} = 2n$, $\underline{\mu} = \frac{4}{n^2}$ and $c_1 = \frac{\bar{\mu}+\underline{\mu}}{\bar{\mu}-\underline{\mu}}$, $c_2 = \frac{2}{\bar{\mu}-\underline{\mu}}$, $\kappa(\mathcal{L}) = \frac{\bar{\mu}}{\underline{\mu}}$
4: $\quad m = \lceil \frac{1}{2}(\sqrt{\kappa(\mathcal{L})}+1)\ln\frac{2}{\epsilon} \rceil$.
5: $\quad [\boldsymbol{\Delta}_0]_r = 0 \quad [\boldsymbol{\Omega}_0]_r = [\boldsymbol{b}_s^i]_r \quad \Theta_0 = 1$.
6: $\quad [\boldsymbol{\Delta}_1]_r = c_2[\boldsymbol{b}_s^i]_r \quad [\boldsymbol{\Omega}_1]_r = [(c_1\boldsymbol{I} - c_2\mathcal{L})\,\boldsymbol{b}_s^i]_r \quad \Theta_1 = c_1$.
7: **for** $\ell = 2$ to $m$ **do**
8: $\quad \Theta_\ell = 2c_1\Theta_{\ell-1} - \Theta_{\ell-2}$.
9: $\quad [\boldsymbol{\Omega}_\ell]_r = [2(c_1\boldsymbol{I} - c_2\mathcal{L})\boldsymbol{\Omega}_{\ell-1}]_r - [\boldsymbol{\Omega}_{\ell-2}]_r$.
10: $\quad [\boldsymbol{\Delta}_\ell]_r = 2c_1[\boldsymbol{\Delta}_{\ell-1}]_r - [\boldsymbol{\Delta}_{\ell-2}]_r + 2c_2[\boldsymbol{\Omega}_{\ell-1}]_r$
11: **end for**
12: Set $[\boldsymbol{d}_s^{(m),i}]_r = \frac{[\boldsymbol{\Delta}_m]_r}{\Theta_m}$

---

## C.2 Convergence Analysis of Distributed Newton Method

Before to proceed to the prove of the Theorem **??** we will establish several intermediate results which are crucial for our convergence analysis. We pose the following assumptions on the local functions:

**Assumption 1.** *The cost functions, $f_r(\cdot) = \sum_{t=1}^{T_i} \mathcal{J}_{MTRL}^{(t)}(\cdot)$, in Equation (**??**) are*

1. *twice continuously differentiable, i.e., $\gamma \boldsymbol{I}_{dk \times dk} \leq \nabla^2 f_r(\cdot) \leq \Gamma \boldsymbol{I}_{dk \times dk}$, with $\gamma$ and $\Gamma$ are constants; and*

2. *Hessian Lipschitz continuous, i.e., $\left\|\nabla^2 f_r(\boldsymbol{x}) - \nabla^2 f_r(\hat{\boldsymbol{x}})\right\|_2 \leq \delta\|\boldsymbol{x} - \hat{\boldsymbol{x}}\|_2$ for all $\boldsymbol{x}, \hat{\boldsymbol{x}} \in \mathbb{R}^{dk}$*

### C.2.1 Primal Dual Transition

Recall, that transition between primal and dual variables is given by a system of differential equations (**??**) and let $\phi_1^{(i)}, \ldots, \phi_{dk}^{(i)}$ denote the solution of this system.

**Lemma** *Let $z_1 = [\mathcal{L}\boldsymbol{\lambda}_1]_r, z_2 = [\mathcal{L}\boldsymbol{\lambda}_2]_r, \ldots, z_{dk} = [\mathcal{L}\boldsymbol{\lambda}_{dk}]_r$. Under Assumption 1, the functions $\phi_1^{(r)}, \ldots, \phi_{dk}^{(r)}$ exhibit bounded partial derivatives with respect to $z_1, \ldots, z_{dk}$. In other words, for any $r = 1, \ldots, dk$:*

$$\left|\frac{\partial \phi_r^{(i)}}{\partial z_1}\right| \leq \frac{\sqrt{dk}}{\gamma}, \quad \cdots, \quad \left|\frac{\partial \phi_r^{(i)}}{\partial z_{dk}}\right| \leq \frac{\sqrt{dk}}{\gamma}$$

*for any $[z_1, \ldots, z_{dk}] \in \mathbb{R}^{dk}$.*

**Proof.** Using the definition of $z_1, \ldots z_{dk}$ we can write:

$$\begin{cases} \frac{\partial f_i}{\partial \phi_1^{(i)}} = -z_1 \\ \frac{\partial f_i}{\partial \phi_2^{(i)}} = -z_2 \\ \vdots \\ \frac{\partial f_i}{\partial \phi_{dk}^{(i)}} = -z_{dk} \end{cases} \tag{8}$$

Taking the derivative with respect to $z_1$ in each equation of system (8) gives:

$$\begin{cases} \frac{\partial^2 f_i}{\partial(\phi_1^{(i)})^2}\frac{\partial\phi_1^{(i)}}{\partial z_1} + \frac{\partial^2 f_i}{\partial\phi_1^{(i)}\partial\phi_2^{(i)}}\frac{\partial\phi_2^{(i)}}{\partial z_1} + \ldots + \frac{\partial^2 f_i}{\partial\phi_1^{(i)}\partial\phi_{dk}^{(i)}}\frac{\partial\phi_{dk}^{(i)}}{\partial z_1} = -1 \\ \frac{\partial^2 f_i}{\partial\phi_2^{(i)}\partial\phi_1^{(i)}}\frac{\partial\phi_1^{(i)}}{\partial z_1} + \frac{\partial^2 f_i}{\partial(\phi_2^{(i)})^2}\frac{\partial\phi_2^{(i)}}{\partial z_1} + \ldots + \frac{\partial^2 f_i}{\partial\phi_2^{(i)}\partial\phi_{dk}^{(i)}}\frac{\partial\phi_{dk}^{(i)}}{\partial z_1} = 0 \\ \vdots \\ \frac{\partial^2 f_i}{\partial\phi_{dk}^{(i)}\partial\phi_1^{(i)}}\frac{\partial\phi_1^{(i)}}{\partial z_1} + \frac{\partial^2 f_i}{\partial\phi_p^{(i)}\partial\phi_2^{(i)}}\frac{\partial\phi_2^{(i)}}{\partial z_1} + \ldots + \frac{\partial^2 f_i}{\partial(\phi_{dk}^{(i)})^2}\frac{\partial\phi_{dk}^{(i)}}{\partial z_1} = 0 \end{cases}$$

Let $\boldsymbol{u}_1 = [\frac{\partial \phi_1^{(i)}}{\partial z_1}, \frac{\partial \phi_2^{(i)}}{\partial z_1}, \ldots, \frac{\partial \phi_{dk}^{(i)}}{\partial z_1}]^\mathsf{T}$ then the above result can be written in matrix vector form:

$$[\nabla^2 f_i]\boldsymbol{u}_1 = -\boldsymbol{e}_1$$

where $\boldsymbol{e}_1 = [1, 0 \ldots, 0] \in \mathbb{R}^{dk}$. Similarly we have:

$$[\nabla^2 f_i]\boldsymbol{u}_2 = -\boldsymbol{e}_2 \quad [\nabla^2 f_i]\boldsymbol{u}_3 = -\boldsymbol{e}_3, \quad \ldots \quad [\nabla^2 f_i]\boldsymbol{u}_{dk} = -\boldsymbol{e}_{dk}$$

with $\boldsymbol{u}_r = [\frac{\partial \phi_1^{(i)}}{\partial z_r}, \frac{\partial \phi_2^{(i)}}{\partial z_r}, \ldots, \frac{\partial \phi_{dk}^{(i)}}{\partial z_r}]^\mathsf{T}$. Combining all these equations gives:

$$[\nabla^2 f_i]\boldsymbol{U} = -\boldsymbol{I}_{dk \times dk} \tag{9}$$

where

$$\boldsymbol{U} = \begin{bmatrix} \frac{\partial \phi_1^{(i)}}{\partial z_1} & \frac{\partial \phi_1^{(i)}}{\partial z_2} & \cdots & \frac{\partial \phi_1^{(i)}}{\partial z_{dk}} \\ \frac{\partial \phi_2^{(i)}}{\partial z_1} & \frac{\partial \phi_2^{(i)}}{\partial z_2} & \cdots & \frac{\partial \phi_2^{(i)}}{\partial z_{dk}} \\ \vdots & & \ddots & \vdots \\ \frac{\partial \phi_{dk}^{(i)}}{\partial z_1} & \frac{\partial \phi_{dk}^{(i)}}{\partial z_2} & \cdots & \frac{\partial \phi_{dk}^{(i)}}{\partial z_{dk}} \end{bmatrix}$$

Notice, Equation (9) implies:

$$\boldsymbol{U} = -[\nabla^2 f_i]^{-1}$$

Hence, using Assumption 1: $||\boldsymbol{U}||_2 \leq \frac{1}{\gamma}$, and:

$$|U_{ij}| \leq ||\boldsymbol{U}||_F \leq \sqrt{dk}||\boldsymbol{U}||_2 \leq \frac{\sqrt{dk}}{\gamma}$$

### C.2.2   Dual Function Properties

In this section we establish important properties of the dual to the problem (**??**).

**Lemma 1.** *The function $q(\boldsymbol{\lambda}) = q(\boldsymbol{\lambda}_1, \ldots, \boldsymbol{\lambda}_{dk})$ abides by the following properties:*

1. *Let $\boldsymbol{y}(\boldsymbol{\lambda})$ be the primal variable corresponding to dual vector $\boldsymbol{\lambda}$. Then the gradient and the Hessian of $q(\boldsymbol{\lambda})$ are given by*

$$\nabla q(\boldsymbol{\lambda}) = \boldsymbol{g}(\boldsymbol{\lambda}) = \boldsymbol{M}\boldsymbol{y}(\boldsymbol{\lambda})$$
$$\nabla^2 q(\boldsymbol{\lambda}) = \boldsymbol{H}(\boldsymbol{\lambda}) = -\boldsymbol{M}\left[\nabla^2 f(\boldsymbol{y}(\boldsymbol{\lambda}))\right]^{-1}\boldsymbol{M}$$

   *where $f(\boldsymbol{y}(\boldsymbol{\lambda})) = \sum_{i=1}^n \sum_{t=1}^{T_i} \mathcal{J}_{MTRL}^{(t)}(\boldsymbol{y}(\boldsymbol{\lambda})))$.*

2. *Denote $\mu_n(\mathcal{L})$ as the largest eigenvalue of the unweighted Laplacian of $\mathbb{G}$ and constants $\delta, \gamma$ are given in Assumption 1. Then, for constant $B = dk\delta \left(\frac{\mu_n(\mathcal{L})}{\gamma}\right)^3$ and for any $\bar{\boldsymbol{\lambda}}, \boldsymbol{\lambda} \in \mathbb{R}^{ndk}$:*

$$||\boldsymbol{H}(\bar{\boldsymbol{\lambda}}) - \boldsymbol{H}(\boldsymbol{\lambda})||_2 \leq B|\bar{\boldsymbol{\lambda}} - \boldsymbol{\lambda}|||_2$$

   *i.e. $\boldsymbol{H}(\boldsymbol{\lambda})$ is Lipschitz continuous with constant $B$.*

**Proof:** To avoid confusion with indexes let us focus on more general optimization problem:

$$\min_{\boldsymbol{x}} f(x) \tag{10}$$
$$s.t. \ \boldsymbol{Ax} = \boldsymbol{b}, \quad \boldsymbol{A} \in \mathbb{R}^{n \times p}, \ \boldsymbol{b} \in \mathbb{R}^n$$

where $f(x)$ twice differentiable strongly convex function, and unknown variable $\boldsymbol{x} \in \mathbb{R}^p$. One can see, that problem (**??**) is a special case of (10) with $f(\cdot) = \sum_{i=1}^n f_i(\cdot)$, $\boldsymbol{A} = \boldsymbol{M}$, and $\boldsymbol{b} = \boldsymbol{0}$[1]. Let $q(\boldsymbol{\lambda})$ be the corresponding dual for (10), with dual variable $\boldsymbol{\lambda} \in \mathbb{R}^n$. We will show that:

$$\nabla^2 q(\boldsymbol{\lambda}) = -\boldsymbol{A}[\nabla^2 f(\boldsymbol{x}(\boldsymbol{\lambda}))]^{-1}\boldsymbol{A}^\mathsf{T} \tag{11}$$
$$\nabla q(\boldsymbol{\lambda}) = \boldsymbol{Ax}(\boldsymbol{\lambda}) - \boldsymbol{b}$$

where $\boldsymbol{x}(\boldsymbol{\lambda}) = argmin_{\boldsymbol{x}} f(\boldsymbol{x}) + \boldsymbol{\lambda}^{\mathsf{T}}(\boldsymbol{A}\boldsymbol{x} - \boldsymbol{b})$ minimizes the Lagrangian of problem (10). Let us denote $\boldsymbol{x}(\boldsymbol{\lambda}) = \boldsymbol{x}^{+}$

$$
\boldsymbol{A} = \begin{bmatrix} a_{11} & \cdots & a_{1p} \\ a_{21} & \cdots & a_{2p} \\ \vdots & \ddots & \vdots \\ a_{n1} & \cdots & a_{np} \end{bmatrix}, \qquad \boldsymbol{x}^{+} = \begin{bmatrix} x_1^{+}(\boldsymbol{\lambda}) \\ x_2^{+}(\boldsymbol{\lambda}) \\ \vdots \\ x_p^{+}(\boldsymbol{\lambda}) \end{bmatrix}, \nabla f(\boldsymbol{x}^{+}) = \begin{bmatrix} z_1(\boldsymbol{x}^{+}) \\ z_2(\boldsymbol{x}^{+}) \\ \vdots \\ z_p(\boldsymbol{x}^{+}) \end{bmatrix} \tag{12}
$$

The optimal primal variable $\boldsymbol{x}(\boldsymbol{\lambda})$ satisfies:

$$
\nabla f(\boldsymbol{x}(\boldsymbol{\lambda})) + \boldsymbol{A}^{\mathsf{T}}\boldsymbol{\lambda} = \boldsymbol{0}. \tag{13}
$$

Using Fenchel's conjugate, the dual function can be written as:

$$
q(\boldsymbol{\lambda}) = -\boldsymbol{b}^{\mathsf{T}}\boldsymbol{\lambda} - f^{*}(-\boldsymbol{A}^{\mathsf{T}}\boldsymbol{\lambda}) \tag{14}
$$

Therefore,

$$
\nabla q(\boldsymbol{\lambda}) = -\boldsymbol{b} - \nabla f^{*}(-\boldsymbol{A}^{\mathsf{T}}\boldsymbol{\lambda}) \tag{15}
$$

Denoting $\boldsymbol{u} = -\boldsymbol{A}^{\mathsf{T}}\boldsymbol{\lambda}$, then the $k^{th}$ component of vector $\nabla f^{*}(-\boldsymbol{A}^{\mathsf{T}}\boldsymbol{\lambda})$ can be written as:

$$
[\nabla f^{*}(-\boldsymbol{A}^{\mathsf{T}}\boldsymbol{\lambda})]_s = \sum_{j=1}^{p} \frac{\partial f^{*}}{\partial u_j} \frac{\partial u_j}{\partial \lambda_k} = -\begin{bmatrix} a_{k1} & a_{k2} & \cdots & a_{kp} \end{bmatrix} \times \begin{bmatrix} \frac{\partial f^{*}}{\partial u_1} \\ \frac{\partial f^{*}}{\partial u_2} \\ \vdots \\ \frac{\partial f^{*}}{\partial u_p} \end{bmatrix}_{-\boldsymbol{A}^{\mathsf{T}}\boldsymbol{\lambda}}
$$

Applying result (13) and the relation between the gradients of function and its Fenchel's conjugate:

$$
\begin{aligned}
\nabla f^{*}(-\boldsymbol{A}^{\mathsf{T}}\boldsymbol{\lambda}) &= -\boldsymbol{A}\nabla_{\boldsymbol{u}} f^{*}(\boldsymbol{u})|_{-\boldsymbol{A}^{\mathsf{T}}\boldsymbol{\lambda}} = -\boldsymbol{A}\nabla_{\boldsymbol{u}} f^{*}(-\boldsymbol{A}^{\mathsf{T}}\boldsymbol{\lambda}) = \\
&- \boldsymbol{A}\nabla_{\boldsymbol{u}} f^{*}(\nabla f(\boldsymbol{x}^{+})) = -\boldsymbol{A}\boldsymbol{x}(\boldsymbol{\lambda})
\end{aligned} \tag{16}
$$

Therefore, the result (15) gives:

$$
\nabla q(\boldsymbol{\lambda}) = -\boldsymbol{b} + \boldsymbol{A}\boldsymbol{x}(\boldsymbol{\lambda}) \tag{17}
$$

which establishes the claim for the dual gradient.
Taking the gradient in (17) gives:

$$
\nabla^2 q(\boldsymbol{\lambda}) = \boldsymbol{A} \underbrace{\begin{bmatrix} \frac{\partial x_1^{+}(\boldsymbol{\lambda})}{\partial \lambda_1} & \frac{\partial x_1^{+}(\boldsymbol{\lambda})}{\partial \lambda_2} & \cdots & \frac{\partial x_1^{+}(\boldsymbol{\lambda})}{\partial \lambda_n} \\ \frac{\partial x_2^{+}(\boldsymbol{\lambda})}{\partial \lambda_1} & \frac{\partial x_2^{+}(\boldsymbol{\lambda})}{\partial \lambda_2} & \cdots & \frac{\partial x_2^{+}(\boldsymbol{\lambda})}{\partial \lambda_n} \\ \vdots & & \ddots & \vdots \\ \frac{\partial x_p^{+}(\boldsymbol{\lambda})}{\partial \lambda_1} & \frac{\partial x_p^{+}(\boldsymbol{\lambda})}{\partial \lambda_2} & \cdots & \frac{\partial x_p^{+}(\boldsymbol{\lambda})}{\partial \lambda_n} \end{bmatrix}}_{\boldsymbol{F}(\boldsymbol{x}^{+})} \tag{18}
$$

Hence, we target matrix $\boldsymbol{F}(\boldsymbol{x}^{+})$ to obtain the form of dual Hessian. Equation (13) gives:

$$
\nabla f(\boldsymbol{x}^{+}) = -\boldsymbol{A}^{\mathsf{T}}\boldsymbol{\lambda}
$$

On the next step, we take partial derivative $\frac{\partial}{\partial \lambda_j}$ for both sides of the above equation for $j = 1, \ldots, n$. For simplicity, consider $\frac{\partial}{\partial \lambda_1}$:

$$\frac{\partial}{\partial \lambda_1} \nabla f(\boldsymbol{x}^+) = \begin{bmatrix} \frac{\partial}{\partial \lambda_1} z_1(\boldsymbol{x}^+) \\ \frac{\partial}{\partial \lambda_1} z_2(\boldsymbol{x}^+) \\ \vdots \\ \frac{\partial}{\partial \lambda_1} z_p(\boldsymbol{x}^+) \end{bmatrix} =$$

$$\begin{bmatrix} \frac{\partial z_1(\boldsymbol{x}^+)}{\partial x_1^+(\boldsymbol{\lambda})} \frac{\partial x_1^+(\boldsymbol{\lambda})}{\partial \lambda_1} + \frac{\partial z_1(\boldsymbol{x}^+)}{\partial x_2^+(\boldsymbol{\lambda})} \frac{\partial x_2^+(\boldsymbol{\lambda})}{\partial \lambda_1} + \cdots + \frac{\partial z_1(\boldsymbol{x}^+)}{\partial x_p^+(\boldsymbol{\lambda})} \frac{\partial x_p^+(\boldsymbol{\lambda})}{\partial \lambda_1} \\ \frac{\partial z_2(\boldsymbol{x}^+)}{\partial x_1^+(\boldsymbol{\lambda})} \frac{\partial x_1^+(\boldsymbol{\lambda})}{\partial \lambda_1} + \frac{\partial z_2(\boldsymbol{x}^+)}{\partial x_2^+(\boldsymbol{\lambda})} \frac{\partial x_2^+(\boldsymbol{\lambda})}{\partial \lambda_1} + \cdots + \frac{\partial z_2(\boldsymbol{x}^+)}{\partial x_p^+(\boldsymbol{\lambda})} \frac{\partial x_p^+(\boldsymbol{\lambda})}{\partial \lambda_1} \\ \vdots \\ \frac{\partial z_p(\boldsymbol{x}^+)}{\partial x_1^+(\boldsymbol{\lambda})} \frac{\partial x_1^+(\boldsymbol{\lambda})}{\partial \lambda_1} + \frac{\partial z_p(\boldsymbol{x}^+)}{\partial x_2^+(\boldsymbol{\lambda})} \frac{\partial x_2^+(\boldsymbol{\lambda})}{\partial \lambda_1} + \cdots + \frac{\partial z_p(\boldsymbol{x}^+)}{\partial x_p^+(\boldsymbol{\lambda})} \frac{\partial x_p^+(\boldsymbol{\lambda})}{\partial \lambda_1} \end{bmatrix} =$$

$$\underbrace{\begin{bmatrix} \frac{\partial z_1(\boldsymbol{x}^+)}{\partial x_1^+(\boldsymbol{\lambda})} & \frac{\partial z_1(\boldsymbol{x}^+)}{\partial x_2^+(\boldsymbol{\lambda})} & \cdots & \frac{\partial z_1(\boldsymbol{x}^+)}{\partial x_p^+(\boldsymbol{\lambda})} \\ \frac{\partial z_2(\boldsymbol{x}^+)}{\partial x_1^+(\boldsymbol{\lambda})} & \frac{\partial z_2(\boldsymbol{x}^+)}{\partial x_2^+(\boldsymbol{\lambda})} & \cdots & \frac{\partial z_2(\boldsymbol{x}^+)}{\partial x_p^+(\boldsymbol{\lambda})} \\ \vdots & & \ddots & \vdots \\ \frac{\partial z_p(\boldsymbol{x}^+)}{\partial x_1^+(\boldsymbol{\lambda})} & \frac{\partial z_p(\boldsymbol{x}^+)}{\partial x_2^+(\boldsymbol{\lambda})} & \cdots & \frac{\partial z_p(\boldsymbol{x}^+)}{\partial x_p^+(\boldsymbol{\lambda})} \end{bmatrix}}_{\nabla^2 f(\boldsymbol{x}^+)} \begin{bmatrix} \frac{\partial x_1^+(\boldsymbol{\lambda})}{\partial \lambda_1} \\ \frac{\partial x_2^+(\boldsymbol{\lambda})}{\partial \lambda_1} \\ \vdots \\ \frac{\partial x_p^+(\boldsymbol{\lambda})}{\partial \lambda_1} \end{bmatrix} = \nabla^2 f(\boldsymbol{x}^+) \begin{bmatrix} \frac{\partial x_1^+(\boldsymbol{\lambda})}{\partial \lambda_1} \\ \frac{\partial x_2^+(\boldsymbol{\lambda})}{\partial \lambda_1} \\ \vdots \\ \frac{\partial x_p^+(\boldsymbol{\lambda})}{\partial \lambda_1} \end{bmatrix} =$$

$$\frac{\partial}{\partial \lambda_1}(-\boldsymbol{A}^\mathsf{T}\boldsymbol{\lambda}) = -\begin{bmatrix} a_{11} \\ a_{12} \\ \vdots \\ a_{1p} \end{bmatrix}$$

Repeating this for $\frac{\partial}{\partial \lambda_2}, \ldots, \frac{\partial}{\partial \lambda_p}$ gives:

$$\nabla^2 f(\boldsymbol{x}^+) \underbrace{\begin{bmatrix} \frac{\partial x_1^+(\boldsymbol{\lambda})}{\partial \lambda_1} & \frac{\partial x_1^+(\boldsymbol{\lambda})}{\partial \lambda_2} & \cdots & \frac{\partial x_1^+(\boldsymbol{\lambda})}{\partial \lambda_n} \\ \frac{\partial x_2^+(\boldsymbol{\lambda})}{\partial \lambda_1} & \frac{\partial x_2^+(\boldsymbol{\lambda})}{\partial \lambda_2} & \cdots & \frac{\partial x_2^+(\boldsymbol{\lambda})}{\partial \lambda_n} \\ \vdots & & \ddots & \vdots \\ \frac{\partial x_p^+(\boldsymbol{\lambda})}{\partial \lambda_1} & \frac{\partial x_p^+(\boldsymbol{\lambda})}{\partial \lambda_2} & \cdots & \frac{\partial x_p^+(\boldsymbol{\lambda})}{\partial \lambda_n} \end{bmatrix}}_{\boldsymbol{F}(\boldsymbol{x}^+)} = -\underbrace{\begin{bmatrix} a_{11} & a_{21} & \cdots & a_{n1} \\ a_{12} & a_{22} & \cdots & a_{n2} \\ \vdots & & \ddots & \vdots \\ a_{1p} & a_{2p} & \cdots & a_{np} \end{bmatrix}}_{\boldsymbol{A}^\mathsf{T}}$$

Therefore, for matrix $\boldsymbol{F}(\boldsymbol{x}^+) = -[\nabla^2 f(\boldsymbol{x}^+)]^{-1} \boldsymbol{A}^\mathsf{T}$ and combining this result with (18) gives:

$$\nabla^2 q(\boldsymbol{\lambda}) = -\boldsymbol{A}[\nabla^2 f(\boldsymbol{x}^+)]^{-1} \boldsymbol{A}^\mathsf{T}$$

Now we are ready to prove the second statement of the lemma. Using $\boldsymbol{M} \preceq \mu_n(\mathcal{L})\boldsymbol{I}$:

$$||[\boldsymbol{H}(\bar{\boldsymbol{\lambda}}) - \boldsymbol{H}(\boldsymbol{\lambda})]\boldsymbol{v}||_2^2 = ||[\boldsymbol{M}([\nabla^2 f(\boldsymbol{y}(\bar{\boldsymbol{\lambda}}))]^{-1} - [\nabla^2 f(\boldsymbol{y}(\boldsymbol{\lambda}))]^{-1})\boldsymbol{M}\boldsymbol{v}]||_2^2 =$$
$$\boldsymbol{v}^\mathsf{T}\boldsymbol{M}([\nabla^2 f(\boldsymbol{y}(\bar{\boldsymbol{\lambda}}))]^{-1} - [\nabla^2 f(\boldsymbol{y}(\boldsymbol{\lambda}))]^{-1})\boldsymbol{M}^2([\nabla^2 f(\boldsymbol{y}(\bar{\boldsymbol{\lambda}}))]^{-1} - [\nabla^2 f(\boldsymbol{y}(\boldsymbol{\lambda}))]^{-1})\boldsymbol{M}\boldsymbol{v} \leq$$
$$\mu_n^2(\mathcal{L})\boldsymbol{v}^\mathsf{T}\boldsymbol{M}([\nabla^2 f(\boldsymbol{y}(\bar{\boldsymbol{\lambda}}))]^{-1} - [\nabla^2 f(\boldsymbol{y}(\boldsymbol{\lambda}))]^{-1})^2\boldsymbol{M}\boldsymbol{v} \leq$$
$$\mu_n^2(\mathcal{L})\mu_{max}^2(|[\nabla^2 f(\boldsymbol{y}(\bar{\boldsymbol{\lambda}}))]^{-1} - [\nabla^2 f(\boldsymbol{y}(\boldsymbol{\lambda}))]^{-1}|)\boldsymbol{v}^\mathsf{T}\boldsymbol{M}^2\boldsymbol{v} \leq$$
$$\mu_n^4(\mathcal{L})\mu_{max}^2(|[\nabla^2 f(\boldsymbol{y}(\bar{\boldsymbol{\lambda}}))]^{-1} - [\nabla^2 f(\boldsymbol{y}(\boldsymbol{\lambda}))]^{-1}|)||\boldsymbol{v}||_2^2$$

Therefore,

$$||\boldsymbol{H}(\bar{\boldsymbol{\lambda}}) - \boldsymbol{H}(\boldsymbol{\lambda})||_2 \leq \mu_n^2(\mathcal{L})\mu_{max}(|[\nabla^2 f(\boldsymbol{y}(\bar{\boldsymbol{\lambda}}))]^{-1} - [\nabla^2 f(\boldsymbol{y}(\boldsymbol{\lambda}))]^{-1}|) \qquad (19)$$

To bound the term $\mu_{max}(|[\nabla^2 f(\boldsymbol{y}(\bar{\boldsymbol{\lambda}}))]^{-1} - [\nabla^2 f(\boldsymbol{y}(\boldsymbol{\lambda}))]^{-1}|)$ we study study the properties of primal Hessian more carefully:

**Claim 2.** *For primal Hessian $\nabla^2 f(\boldsymbol{y}(\boldsymbol{\lambda}))$ the following properties are true*

$$\gamma \preceq \nabla^2 f(\boldsymbol{y}(\boldsymbol{\lambda})) \preceq \Gamma \tag{20}$$

$$\mu_{max}(|[\nabla^2 f(\boldsymbol{y}(\bar{\boldsymbol{\lambda}}))]^{-1} - [\nabla^2 f(\boldsymbol{y}(\boldsymbol{\lambda}))]^{-1}|) \leq \tag{21}$$

$$\delta \max_{i \in \mathbb{V}} \sqrt{\sum_{k=1}^{p} \left([\boldsymbol{y}_s]_i(\bar{\boldsymbol{\lambda}}) - [\boldsymbol{y}_s]_i(\boldsymbol{\lambda})\right)^2}$$

*for any $\bar{\boldsymbol{\lambda}}, \boldsymbol{\lambda} \in \mathbb{R}^p$.*

**Proof.** Firstly, notice that for any $j \neq i$ and any $r = 1 \ldots, p$:

$$\frac{\partial^2 f}{\partial[\boldsymbol{y}_1]_i \partial[\boldsymbol{y}_r]_j} = \frac{\partial^2 f}{\partial[\boldsymbol{y}_2]_i \partial[\boldsymbol{y}_r]_j} = \ldots = \frac{\partial^2 f}{\partial[\boldsymbol{y}_p]_i \partial[\boldsymbol{y}_r]_j} = 0$$

Hence, the sparsity pattern of primal Hessian allows the symmetric reordering of rows and columns such that $\nabla^2 f(\boldsymbol{y}(\boldsymbol{\lambda}))$ is transformed into the block diagonal matrix:

$$\boldsymbol{W}(\boldsymbol{\lambda}) = \begin{bmatrix} \nabla^2 f_1(\boldsymbol{\lambda}) & \mathbf{0} & \cdots & \mathbf{0} \\ \mathbf{0} & \nabla^2 f_2(\boldsymbol{\lambda}) & \cdots & \mathbf{0} \\ \vdots & & \ddots & \vdots \\ \mathbf{0} & \mathbf{0} & \cdots & \nabla^2 f_p(\boldsymbol{\lambda}) \end{bmatrix}$$

The matrix $\boldsymbol{W}(\boldsymbol{\lambda})$ preserves the important properties of $\nabla^2 f(\boldsymbol{y}(\boldsymbol{\lambda}))$. Particularly, the spectrum of these two matrices are the same. Indeed, let $\boldsymbol{T}_{ij}$ is the operator that swaps $i^{th}$ and $j^{th}$ rows of some arbitrary matrix $\boldsymbol{A}$ and let $\bar{\boldsymbol{A}}$ be the result of such transformation. Then, $\bar{\boldsymbol{A}} = \boldsymbol{T}_{ij} \boldsymbol{A} \boldsymbol{T}_{ij}$, and using $\boldsymbol{T}_{ij}^2 = \boldsymbol{I}$:

$$det(\bar{\boldsymbol{A}} - \mu\boldsymbol{I}) = det(\boldsymbol{T}_{ij}\boldsymbol{A}\boldsymbol{T}_{ij} - \mu\boldsymbol{I}) = det(\boldsymbol{T}_{ij}(\boldsymbol{A} - \mu\boldsymbol{I})\boldsymbol{T}_{ij}) =$$
$$det(\boldsymbol{A} - \mu\boldsymbol{I})det(\boldsymbol{T}_{ij}^2) = det(\boldsymbol{A} - \mu\boldsymbol{I})$$

Since $\boldsymbol{W}(\boldsymbol{\lambda})$ is constructed from $\nabla^2 f(\boldsymbol{y}(\boldsymbol{\lambda}))$ by symmetric reordering rows and columns, then $Spectrum(\boldsymbol{W}(\boldsymbol{\lambda})) = Spectrum(\nabla^2 f(\boldsymbol{y}(\boldsymbol{\lambda})))$. Using the Assumption 1 it implies:

$$\gamma \preceq \boldsymbol{W}(\boldsymbol{\lambda}) \preceq \Gamma$$

To prove (21), notice that if $\bar{\boldsymbol{A}} = \boldsymbol{T}_{ij} \boldsymbol{A} \boldsymbol{T}_{ij}$ and $\boldsymbol{A}$ is invertible, then so $\bar{\boldsymbol{A}}$ and using $\boldsymbol{T}_{ij}^{-1} = \boldsymbol{T}_{ij}$:

$$det(\bar{\boldsymbol{A}}^{-1} - \mu\boldsymbol{I}) = det(\boldsymbol{T}_{ij}^{-1}\boldsymbol{A}^{-1}\boldsymbol{T}_{ij}^{-1} - \mu\boldsymbol{I}) = det(\boldsymbol{T}_{ij}(\boldsymbol{A}^{-1} - \mu\boldsymbol{I})\boldsymbol{T}_{ij}) =$$
$$det(\boldsymbol{A}^{-1} - \mu\boldsymbol{I})$$

Denote $\{\boldsymbol{T}_1, \ldots, \boldsymbol{T}_l\}$ is a collection of operators that swaps the rows of matrix $\nabla^2 f(\boldsymbol{y}(\boldsymbol{\lambda}))$ to transform it to $\boldsymbol{W}(\boldsymbol{\lambda})$, i.e.

$$\boldsymbol{W}(\boldsymbol{\lambda}) = \boldsymbol{T}_1 \cdots \boldsymbol{T}_l \nabla^2 f(\boldsymbol{y}(\boldsymbol{\lambda})) \boldsymbol{T}_l \cdots \boldsymbol{T}_1$$

Then $[\nabla^2 f(\boldsymbol{y}(\boldsymbol{\lambda}))]^{-1} = \boldsymbol{T}_l \cdots \boldsymbol{T}_1 \boldsymbol{W}^{-1}(\boldsymbol{\lambda}) \boldsymbol{T}_1 \cdots \boldsymbol{T}_l$, and using the Assumption 1:

$$\mu_{max}(||[\nabla^2 f(\boldsymbol{y}(\bar{\boldsymbol{\lambda}}))]^{-1} - [\nabla^2 f(\boldsymbol{y}(\boldsymbol{\lambda}))]^{-1}||) =$$
$$\mu_{max}(\boldsymbol{T}_l \cdots \boldsymbol{T}_1 |\boldsymbol{W}^{-1}(\bar{\boldsymbol{\lambda}}) - \boldsymbol{W}^{-1}(\boldsymbol{\lambda})| \boldsymbol{T}_1 \cdots \boldsymbol{T}_l) \leq \mu_{max}(|\boldsymbol{W}^{-1}(\bar{\boldsymbol{\lambda}}) - \boldsymbol{W}^{-1}(\boldsymbol{\lambda})|) \leq$$
$$\max_{i \in \mathbb{V}} \mu_{max}(|[\nabla^2 f_i([\boldsymbol{y}_1]_i(\bar{\boldsymbol{\lambda}}), \ldots [\boldsymbol{y}_p]_i(\bar{\boldsymbol{\lambda}}))]^{-1} - [\nabla^2 f_i([\boldsymbol{y}_1]_i(\boldsymbol{\lambda}), \ldots [\boldsymbol{y}_p]_i(\boldsymbol{\lambda}))]^{-1}|) =$$
$$\max_{i \in \mathbb{V}} ||[\nabla^2 f_i([\boldsymbol{y}_1]_i(\bar{\boldsymbol{\lambda}}), \ldots [\boldsymbol{y}_p]_i(\bar{\boldsymbol{\lambda}}))]^{-1} - [\nabla^2 f_i([\boldsymbol{y}_1]_i(\boldsymbol{\lambda}), \ldots [\boldsymbol{y}_p]_i(\boldsymbol{\lambda}))]^{-1}||_2 \leq$$
$$\frac{\delta}{\gamma^2} \max_{i \in \mathbb{V}} ||([\boldsymbol{y}_1]_i(\bar{\boldsymbol{\lambda}}), \ldots [\boldsymbol{y}_p]_i(\bar{\boldsymbol{\lambda}})) - ([\boldsymbol{y}_1]_i(\boldsymbol{\lambda}), \ldots [\boldsymbol{y}_p]_i(\boldsymbol{\lambda}))||_2 =$$
$$\frac{\delta}{\gamma^2} \max_{i \in \mathbb{V}} \sqrt{\sum_{k=1}^{p} \left([\boldsymbol{y}_s]_i(\bar{\boldsymbol{\lambda}}) - [\boldsymbol{y}_s]_i(\boldsymbol{\lambda})\right)^2}$$

which establishes (21).

Consider the term $\big([\boldsymbol{y}_s]_i(\bar{\boldsymbol{\lambda}}) - [\boldsymbol{y}_s]_i(\boldsymbol{\lambda})\big)$. Using the above results we can write:

$$\big|y_k(i)(\bar{\boldsymbol{\lambda}}) - y_k(i)(\boldsymbol{\lambda})\big| = |\phi_s^{(i)}([\mathcal{L}\bar{\boldsymbol{\lambda}}_1]_i, \ldots, [\mathcal{L}\bar{\boldsymbol{\lambda}}_p]_i) - \phi_s^i([\mathcal{L}\boldsymbol{\lambda}_1]_i, \ldots, [\mathcal{L}\boldsymbol{\lambda}_p]_i)| \le$$

$$\frac{\sqrt{p}}{\gamma}\sqrt{\sum_{r=1}^{p}\big([\mathcal{L}\bar{\boldsymbol{\lambda}}_r]_i - [\mathcal{L}\boldsymbol{\lambda}_r]_i\big)^2} = \frac{\sqrt{p}}{\gamma}\sqrt{\sum_{r=1}^{p}\big[\boldsymbol{L}_{\mathcal{G}}(\bar{\boldsymbol{\lambda}}_r - \boldsymbol{\lambda}_r)\big]_i^2} \le \frac{\sqrt{p}}{\gamma}\sqrt{\sum_{r=1}^{p}||\mathcal{L}(\bar{\boldsymbol{\lambda}}_r - \boldsymbol{\lambda}_r)||_2^2} =$$

$$\frac{\sqrt{p}}{\gamma}\sqrt{\sum_{r=1}^{p}(\bar{\boldsymbol{\lambda}}_r - \boldsymbol{\lambda}_r)^{\mathsf{T}}\mathcal{L}^2(\bar{\boldsymbol{\lambda}}_r - \boldsymbol{\lambda}_r)} \le \frac{\sqrt{p}}{\gamma}\sqrt{\mu_n^2(\mathcal{L})\sum_{r=1}^{p}(\bar{\boldsymbol{\lambda}}_r - \boldsymbol{\lambda}_r)^{\mathsf{T}}(\bar{\boldsymbol{\lambda}}_r - \boldsymbol{\lambda}_r)} =$$

$$= \mu_n(\mathcal{L})\frac{\sqrt{p}}{\gamma}||\bar{\boldsymbol{\lambda}} - \boldsymbol{\lambda}||_2$$

where

$$\sum_{r=1}^{p}(\bar{\boldsymbol{\lambda}}_r - \boldsymbol{\lambda}_r)^{\mathsf{T}}(\bar{\boldsymbol{\lambda}}_r - \boldsymbol{\lambda}_r) = ||\bar{\boldsymbol{\lambda}} - \boldsymbol{\lambda}||_2^2$$

is used. Hence,

$$\big([\boldsymbol{y}_s]_i(\bar{\boldsymbol{\lambda}}) - [\boldsymbol{y}_s]_i(\boldsymbol{\lambda})\big)^2 \le \mu_n^2(\mathcal{L})\frac{p}{\gamma^2}||\bar{\boldsymbol{\lambda}} - \boldsymbol{\lambda}||_2^2$$

Combining this result with (21) gives:

$$\mu_{max}(||[\nabla^2 f(\boldsymbol{y}(\bar{\boldsymbol{\lambda}}))]^{-1} - [\nabla^2 f(\boldsymbol{y}(\boldsymbol{\lambda}))]^{-1}||) \le \frac{\delta}{\gamma^2}\mu_n(\mathcal{L})\frac{p}{\gamma}||\bar{\boldsymbol{\lambda}} - \boldsymbol{\lambda}||_2$$

and applying it to (19) gives:

$$||\boldsymbol{H}(\bar{\boldsymbol{\lambda}}) - \boldsymbol{H}(\boldsymbol{\lambda})||_2 \le \delta p\left(\frac{\mu_n(\mathcal{L})}{\gamma}\right)^3||\bar{\boldsymbol{\lambda}} - \boldsymbol{\lambda}||_{\mathbf{2}} = \delta dk\left(\frac{\mu_n(\mathcal{L})}{\gamma}\right)^3||\bar{\boldsymbol{\lambda}} - \boldsymbol{\lambda}||_{\mathbf{2}} = B||\bar{\boldsymbol{\lambda}} - \boldsymbol{\lambda}||_2$$

In other words, dual Hessian is Lipschitz continuous with constant $B = \delta dk\left(\frac{\mu_n(\mathcal{L})}{\gamma}\right)^3$

### C.2.3   Dual Gradient Bounds

The following Lemma studies the change of the norm of dual gradient for Distributed Newton iteration scheme and plays a crucial role for the convergence analysis:

**Lemma 2.** *Let us consider iteration scheme given by $\boldsymbol{\lambda}_{s+1} = \boldsymbol{\lambda}_s + \alpha_s \boldsymbol{d}_s^{(m)}$ and denote*

$$\boldsymbol{\epsilon}_s = \boldsymbol{H}_s \boldsymbol{d}_s^{(m)} + \boldsymbol{g}_s$$

*be the approximation error vector corresponding to $\epsilon-$ approximated Newton direction vector $\boldsymbol{d}_s^{(m)}$ and $\boldsymbol{g}_s = \boldsymbol{g}(\boldsymbol{\lambda}_s) = \nabla q(\boldsymbol{\lambda}_s)$. Then for any $\alpha_k \in (0,1]$*

$$||\boldsymbol{g}_{s+1}||_2 \le \tag{22}$$

$$(1 - \alpha_k)||\boldsymbol{g}_s||_2 + \alpha_k^2 B\frac{\Gamma^2}{\mu_2^4(\mathcal{L})}||\boldsymbol{g}_s||_2^2 + \alpha_k||\boldsymbol{\epsilon}_s||_2 + \alpha_s^2 B\frac{\Gamma^2}{\mu_2^4(\mathcal{L})}||\boldsymbol{\epsilon}_s||_2^2$$

*where $B$ is defined in Lemma 1 and $\mu_2(\mathcal{L})$ is the smallest nonzero eigenvalue of unweighted Laplacian of $\mathcal{G}$.*

**Proof**. Using definition of $\boldsymbol{\epsilon}_s$ for the dual gradient we have:

$$\boldsymbol{g}(\boldsymbol{\lambda}_s + \alpha_s \boldsymbol{d}_s^{(m)}) = \boldsymbol{g}(\boldsymbol{\lambda}_s) + \int_0^1 \boldsymbol{H}(\boldsymbol{\lambda}_s + t\alpha_s \boldsymbol{d}_s^{(m)})\alpha_s \boldsymbol{d}_s^{(m)}dt =$$

$$\boldsymbol{g}(\boldsymbol{\lambda}_s) + \int_0^1 \big[\boldsymbol{H}(\boldsymbol{\lambda}_s + t\alpha_s \boldsymbol{d}_s^{(m)}) - \boldsymbol{H}(\boldsymbol{\lambda}_s)\big]\alpha_s \boldsymbol{d}_s^{(m)}dt + \alpha_s \int_0^1 \boldsymbol{H}(\boldsymbol{\lambda}_s)\boldsymbol{d}_s^{(m)}dt =$$

$$\boldsymbol{g}(\boldsymbol{\lambda}_s) + \int_0^1 \big[\boldsymbol{H}(\boldsymbol{\lambda}_s + t\alpha_s \boldsymbol{d}_s^{(m)}) - \boldsymbol{H}(\boldsymbol{\lambda}_s)\big]\alpha_s \boldsymbol{d}_s^{(m)}dt + \alpha_s(\boldsymbol{\epsilon}_s - \boldsymbol{g}(\boldsymbol{\lambda}_s))$$

Applying $\boldsymbol{g}_{s+1} = \boldsymbol{g}(\boldsymbol{\lambda}_s + \alpha_s \boldsymbol{d}_s^{(m)})$, $\boldsymbol{g}_s = \boldsymbol{g}(\boldsymbol{\lambda}_s)$ and Lemma 1:

$$||\boldsymbol{g}_{s+1}||_2 \leq (1 - \alpha_s)||\boldsymbol{g}_s||_2 + \alpha_s||\boldsymbol{\epsilon}_s||_2 + \frac{1}{2}\alpha_k^2 B||\boldsymbol{d}_s^{(m)}||_2^2 =$$

$$(1 - \alpha_s)||\boldsymbol{g}_s||_2 + \alpha_s||\boldsymbol{\epsilon}_s||_2 + \frac{1}{2}\alpha_s^2 B||\boldsymbol{H}^\dagger(\boldsymbol{\lambda}_s)(\boldsymbol{g}_s - \boldsymbol{\epsilon}_s)||_2^2 \leq$$

$$(1 - \alpha_s)||\boldsymbol{g}_s||_2 + \alpha_s||\boldsymbol{\epsilon}_s||_2 + \alpha_s^2 B||\boldsymbol{H}^\dagger(\boldsymbol{\lambda}_s)||_2^2 (||\boldsymbol{g}_s||_2^2 + ||\boldsymbol{\epsilon}_s||_2^2)$$

Investigating the explicit form of dual Hessian gives $||\boldsymbol{H}^\dagger(\boldsymbol{\lambda}_s)||_2 \leq \frac{\Gamma}{\mu_2^2(\mathcal{L})}$. Hence,

$$||\boldsymbol{g}_{s+1}||_2 \leq (1 - \alpha_s)||\boldsymbol{g}_s||_2 + \alpha_s^2 B \frac{\Gamma^2}{\mu_2^4(\mathcal{L})}||\boldsymbol{g}_s||_2^2 + \alpha_s||\boldsymbol{\epsilon}_s||_2 + \alpha_s^2 B \frac{\Gamma^2}{\mu_2^4(\mathcal{L})}||\boldsymbol{\epsilon}_s||_2^2.$$

### C.2.4 Newton Method Proofs

Similar to centralized Newton method, the step size $\alpha_s$ in iteration scheme $\boldsymbol{\lambda}_{s+1} = \boldsymbol{\lambda}_s + \alpha_s \boldsymbol{d}_s^{(m)}$ should be chosen carefully in order to attain quadratic convergence. In this part of the Appendix, we consider the distributed version of Armijo rule is given in Algorithm 3. We use $\boldsymbol{g} = \boldsymbol{M}\boldsymbol{y} =$

---

**Algorithm 3** : **Distributed Line Search**

---

    **Input:** The constants $\sigma \in \left(0, \frac{1}{2}\right]$ and $\beta \in (0, 1)$, parameters $\epsilon, \Gamma, \gamma, \delta$. The $i^{th}$ component of dual gradient chunks: $\{[\mathcal{L}\boldsymbol{y}_r]_i\}_{r=1}^{dk}$
    **Output:** step size $\alpha_s$.
    Set $m_i = 0$.
    Compute $\eta_i = \max_r\{|[\mathcal{L}\boldsymbol{y}_r]_i|\}$.
    Compute $\max_i\{\eta_i\}$ using maximal consensus protocol.
    **while** $\max_r\{|[\mathcal{L}\boldsymbol{y}_r]_i|\} > (1 - \sigma\beta^{m_i})\sqrt{n}\max_i\{\eta_i\} + 2\epsilon\frac{n\gamma^2}{dk\delta\Gamma}$ **do**
        $m_i = m_i + 1$.
    **end while**
    Compute $\hat{m} = \max_i\{m_i\}$ using maximal consensus protocol.
    Set $\alpha_s = \beta^{\hat{m}}$.

---

$\left(\left(\mathcal{L}\boldsymbol{y}_1\right)^\mathsf{T}, \ldots, \left(\mathcal{L}\boldsymbol{y}_{dk}\right)^\mathsf{T}\right)^\mathsf{T}$. Algorithm 3 requires only $\mathcal{O}(diam(\mathcal{G}))$ time steps and conducts only exact computations. The following Lemma studies the change of step size given by the proposed backtracking line search procedure:

**Lemma 3.** *Let step size $\alpha_s$ is chosen according to Algorithm 3 and let $\boldsymbol{g}_s$ be the dual gradient evaluated at $\boldsymbol{\lambda}_s$. Then*

    *1. If $||\boldsymbol{g}_s||_2 \leq \frac{\mu_2^4(\mathcal{L})}{2B\Gamma^2}$ then $\alpha_s = 1$*

    *2. If $||\boldsymbol{g}_s||_2 > \frac{\mu_2^4(\mathcal{L})}{2B\Gamma^2}$ then $\alpha_s \geq \beta\frac{\mu_2^4(\mathcal{L})}{2B\Gamma^2 \max_i\{\eta_i\}}$*

*where $B$ is a constant defined in Lemma 1 and $\mu_2(\mathcal{L}), \mu_n(\mathcal{L})$ are the smallest and largest nonzero eigenvalues of the unweighted Laplacian of $\mathcal{G}$.*

**Proof.** Combining $||\boldsymbol{g}_s||_2 \leq \frac{\mu_2^4(\mathcal{L})}{2B\Gamma^2}$ with Lemma 2 implies:

$$||\boldsymbol{g}_{s+1}||_2 \leq \left(\frac{3}{2} - \alpha_s\right)||\boldsymbol{g}_s||_2 + \alpha_s||\boldsymbol{\epsilon}_s||_2 + \alpha_s^2 B\frac{\Gamma^2}{\mu_2^4(\mathcal{L})}||\boldsymbol{\epsilon}_s||_2^2$$

Since $||\boldsymbol{\epsilon}_s||_2 \leq \epsilon \frac{\mu_n(\mathcal{L})}{\mu_n(\mathcal{L})} \sqrt{\frac{\Gamma}{\gamma}} ||\boldsymbol{g}_s||_2$, $||\boldsymbol{g}_s||_2 \leq \frac{\mu_2^4(\mathcal{L})}{2B\Gamma^2}$ and $\alpha_s \leq 1$, then

$$||\boldsymbol{g}_{s+1}||_2 \leq \left(\frac{3}{2} - \alpha_s\right) ||\boldsymbol{g}_s||_2 + \alpha_s \epsilon \frac{\mu_n(\mathcal{L})}{\mu_n(\mathcal{L})} \sqrt{\frac{\Gamma}{\gamma}} ||\boldsymbol{g}_s||_2 + \alpha_s^2 \epsilon^2 B \frac{\Gamma^3}{\mu_2^6(\mathcal{L})} \frac{\mu_n^2(\mathcal{L})}{\gamma} ||\boldsymbol{g}_s||_2^2 \leq$$

$$\left(\frac{3}{2} - \alpha_s\right) ||\boldsymbol{g}_s||_2 + \epsilon \frac{\mu_n(\mathcal{L})}{\mu_n(\mathcal{L})} \sqrt{\frac{\Gamma}{\gamma}} ||\boldsymbol{g}_s||_2 + \epsilon^2 B \frac{\Gamma^3}{\mu_2^6(\mathcal{L})} \frac{\mu_n^2(\mathcal{L})}{\gamma} ||\boldsymbol{g}_s||_2^2 \leq$$

$$\left(\frac{3}{2} - \alpha_s\right) ||\boldsymbol{g}_s||_2 + \frac{1}{2}\epsilon \frac{\mu_2^2(\mathcal{L})\gamma^2}{\mu_n(\mathcal{L})dk\Gamma\delta} \left[\frac{\mu_2(\mathcal{L})}{\mu_n(\mathcal{L})} \sqrt{\frac{\gamma}{\Gamma}} + \frac{\epsilon}{2}\right] = \left(\frac{3}{2} - \alpha_s\right) ||\boldsymbol{g}_s||_2 + \hat{B}$$

where we denote $\hat{B} = \frac{1}{2}\epsilon \frac{\mu_2^2(\mathcal{L})\gamma^2}{\mu_n(\mathcal{L})dk\Gamma\delta} \left[\frac{\mu_2(\mathcal{L})}{\mu_n(\mathcal{L})} \sqrt{\frac{\gamma}{\Gamma}} + \frac{\epsilon}{2}\right] \leq 2\epsilon \frac{n\gamma^2}{dk\Gamma\delta}$ for $\epsilon \leq \frac{4}{n^3}\sqrt{\frac{\gamma}{\Gamma}}$. Since $||\boldsymbol{g}_{s+1}||_2 \geq \max_r\{|[\mathcal{L}\boldsymbol{y}_r]_i|\}$ and $||\boldsymbol{g}_s||_2 \leq \sqrt{n} \max_i\{\eta_i\}$, then

$$\max_r\{|[\mathcal{L}\boldsymbol{y}_r]_i|\} \leq \left(\frac{3}{2} - \alpha_s\right) \sqrt{n} \max_i\{\eta_i\} + 2\epsilon \frac{n\gamma^2}{dk\Gamma\delta}$$

Notice that if $m_i = 0$ the $\frac{3}{2} - \beta^{m_i} \leq 1 - \sigma\beta^{m_i}$ Therefore, for $m_i = 0$ we have

$$\max_r\{|[\mathcal{L}\boldsymbol{y}_r]_i|\} \leq \left(1 - \sigma\beta^{m_i}\right) \sqrt{n} \max_i\{\eta_i\} + 2\epsilon \frac{n\gamma^2}{dk\Gamma\delta}$$

In other words, Algorithm 3 returns $\alpha_s = \beta^0 = 1$.

For the case $||\boldsymbol{g}_s||_2 > \frac{\mu_2^4(\mathcal{L})}{2B\Gamma^2}$ consider $\bar{\alpha}_s = \frac{\mu_2^4(\mathcal{L})}{2B\Gamma^2\sqrt{n}\max_i\{\eta_i\}}$. Because $||\boldsymbol{g}_s||_2 \leq \sqrt{n} \max_i\{\eta_i\}$ and $||\boldsymbol{g}_s||_2 > \frac{\mu_2^4(\mathcal{L})}{2B\Gamma^2}$ then $\bar{\alpha}_s < 1$. Hence, applying $\bar{\alpha}_s$ with $\epsilon \leq \frac{4}{n^3}\sqrt{\frac{\gamma}{\Gamma}}$ for (22) gives:

$$||\boldsymbol{g}_{s+1}||_2 \leq (1 - \bar{\alpha}_s)||\boldsymbol{g}_s||_2 + \bar{\alpha}_s^2 B \frac{\Gamma^2}{\mu_2^4(\mathcal{L})} ||\boldsymbol{g}_s||_2^2 + \bar{\alpha}_s ||\boldsymbol{\epsilon}_s||_2 + \bar{\alpha}_s^2 B \frac{\Gamma^2}{\mu_2^4(\mathcal{L})} ||\boldsymbol{\epsilon}_s||_2^2 =$$

$$||\boldsymbol{g}_s||_2 + \bar{\alpha}_s ||\boldsymbol{\epsilon}_s||_2 + \bar{\alpha}_s^2 B \frac{\Gamma^2}{\mu_2^4(\mathcal{L})} ||\boldsymbol{\epsilon}_s||_2^2 - \bar{\alpha}_s ||\boldsymbol{g}_s||_2 \left[1 - \bar{\alpha}_s B \frac{\Gamma^2}{\mu_2^4(\mathcal{L})} ||\boldsymbol{g}_s||_2\right] \leq$$

$$||\boldsymbol{g}_s||_2 + \bar{\alpha}_s \epsilon \frac{\mu_n(\mathcal{L})}{\mu_n(\mathcal{L})} \sqrt{\frac{\Gamma}{\gamma}} ||\boldsymbol{g}_s||_2 + \bar{\alpha}_s^2 \epsilon^2 B \frac{\Gamma^2}{\mu_2^4(\mathcal{L})} \frac{\mu_n^2(\mathcal{L})}{\mu_2^2(\mathcal{L})} \frac{\Gamma}{\gamma} ||\boldsymbol{g}_s||_2^2 -$$

$$\bar{\alpha}_s ||\boldsymbol{g}_s||_2 \left[1 - \frac{||\boldsymbol{g}_s||_2}{2\sqrt{n}\max_i\{\eta_i\}}\right] \leq ||\boldsymbol{g}_s||_2 + \bar{\alpha}_s \epsilon \frac{\mu_n(\mathcal{L})}{\mu_n(\mathcal{L})} \sqrt{\frac{\Gamma}{\gamma}} ||\boldsymbol{g}_s||_2 +$$

$$\bar{\alpha}_s^2 \epsilon^2 B \frac{\Gamma^2}{\mu_2^4(\mathcal{L})} \frac{\mu_n^2(\mathcal{L})}{)} \mu_2^2(\mathcal{L}) \frac{\Gamma}{\gamma} ||\boldsymbol{g}_s||_2^2 - \frac{1}{2}\bar{\alpha}_s ||\boldsymbol{g}_s||_2 = \left(1 - \frac{\bar{\alpha}_s}{2}\right) ||\boldsymbol{g}_s||_2 +$$

$$\epsilon \frac{\mu_n(\mathcal{L})}{\mu_n(\mathcal{L})} \sqrt{\frac{\Gamma}{\gamma}} ||\boldsymbol{g}_s||_2 \frac{||\boldsymbol{g}_s||_2}{\frac{2B\Gamma^2}{\mu_2^2(\mathcal{L})}\sqrt{n}\max_i\{\eta_i\}} + \epsilon^2 \frac{\mu_n^2(\mathcal{L})}{\mu_2^2(\mathcal{L})} \frac{\Gamma}{\gamma} \frac{1}{4\frac{B\Gamma^2}{\mu_2^4(\mathcal{L})}} \frac{||\boldsymbol{g}_s||_2^2}{n\max_i\{\eta_i^2\}} \leq$$

$$\left(1 - \frac{\bar{\alpha}_s}{2}\right) ||\boldsymbol{g}_s||_2 + \hat{B} \leq \left(1 - \frac{\bar{\alpha}_s}{2}\right) ||\boldsymbol{g}_s||_2 + 2\epsilon \frac{n\gamma^2}{dk\delta\Gamma}$$

In other words, we establishes:

$$||\boldsymbol{g}_{s+1}||_2 \leq (1 - \sigma\bar{\alpha}_s)||\boldsymbol{g}_s||_2 + 2\epsilon \frac{n\gamma^2}{dk\delta\Gamma}$$

Applying again $||\boldsymbol{g}_{s+1}||_2 \geq \max_r\{|[\mathcal{L}\boldsymbol{y}_r]_i|\}$ and $||\boldsymbol{g}_s||_2 \leq \sqrt{n} \max_i\{\eta_i\}$ gives:

$$\max_r\{|[\mathcal{L}\boldsymbol{y}_r]_i|\} \leq (1 - \sigma\bar{\alpha}_s) \sqrt{n} \max_i\{\eta_i\} + 2\epsilon \frac{n\gamma^2}{dk\Gamma\delta}$$

Therefore, Algorithm 3 returns $\alpha_s \geq \beta\bar{\alpha}_s = \beta \frac{\mu_2^4(\mathcal{L})}{2B\Gamma^2 \max_i\{\eta_i\}}$.

### C.2.5 Proof of the Main Theorem

In this appendix we prove Theorem **??**:

**Theorem** *Let $\gamma$, $\Gamma$, $\delta$ $B$ be the constants defined in Assumption 1 and Lemma 1, $\mu_2(\mathcal{L})$ and $\mu_n(\mathcal{L})$ representing the smallest and largest nonzero eigenvalues of the unweighted Laplacian of $\mathcal{G}$, $\epsilon \leq \frac{\beta}{8} \frac{\gamma^3}{\Gamma^2 p \delta} \frac{\mu_2^4(\mathcal{L})}{\mu_n^3(\mathcal{L})}$ be the precision parameter for the SDD solver. Consider our iteration scheme with the step size $\alpha_s$ is calculated by Algorithm 3. Then, this iteration scheme exhibits two convergence phases:*

1. ***Strict Decreases Phase** If $\|g_s\|_2 > \frac{\mu_2^4(\mathcal{L})}{2B\Gamma^2}$, then*

$$\|g_{s+1}\|_2 - \|g_k\|_2 \leq -\frac{\beta}{8\sqrt{n}p\delta} \frac{\gamma^3}{\Gamma^2} \frac{\mu_2^4(\mathcal{L})}{\mu_n^3(\mathcal{L})}$$

   *where parameter $\beta \in (0,1)$.*

2. ***Quadratic Decreases Phase** If $\|g_s\|_2 \leq \frac{\mu_2^4(\mathcal{L})}{2B\Gamma^2}$, then for any $o \geq 1$:*

$$\|g_{s+o}\|_2 \leq \frac{1}{2^{2o}\frac{B\Gamma^2}{\mu_2^4(\mathcal{L})}} + \hat{B} + \frac{\tilde{\Lambda}}{\frac{B\Gamma^2}{\mu_2^4(\mathcal{L})}} \left[\frac{2^{2^l-1}-1}{2^{2^l}}\right]$$

   *where*

$$\hat{B} = \frac{1}{2}\epsilon \frac{\mu_2^2(\mathcal{L})\gamma^2}{\mu_n(\mathcal{L})p\Gamma\delta}\left[\frac{\mu_2(\mathcal{L})}{\mu_n(\mathcal{L})}\sqrt{\frac{\gamma}{\Gamma}} + \frac{\epsilon}{2}\right] \sim \mathcal{O}(\epsilon)$$

$$\tilde{\Lambda} = \hat{B}\frac{4B\Gamma^2}{\mu_2^4(\mathcal{L})}\left[1 + \hat{B}\frac{B\Gamma^2}{\mu_2^4(\mathcal{L})}\right] \sim \mathcal{O}(\epsilon)$$

**Proof.** *We will proof the above theorem by handling each of the cases separately. We start by considering the case when $\|g_s\|_2 > \frac{\mu_2^4(\mathcal{L})}{2B\Gamma^2}$. Then, according to Lemma 3: $\alpha_s \geq \beta\frac{\mu_2^4(\mathcal{L})}{2B\Gamma^2 \max_i\{\eta_i\}}$ and Equation (22) we have:*

$$\|g_{s+1}\|_2 \leq (1 - \frac{1}{2}\beta\bar{\alpha}_s)\|g_s\|_2 + 2\epsilon\frac{n\gamma^2}{p\delta\Gamma}$$

*Choosing $\epsilon \leq \frac{\beta}{8}\frac{\gamma^3}{\Gamma^2 p\delta}\frac{\mu_2^4(\mathcal{L})}{\mu_n^3(\mathcal{L})}$ implies $2\epsilon\frac{n\gamma^2}{p\delta\Gamma} \leq \frac{1}{4}\beta\bar{\alpha}_s\|g_s\|_2$ and*

$$\|g_{s+1}\|_2 - \|g_s\|_2 \leq -\frac{1}{4}\beta\bar{\alpha}_s\|g_s\|_2 \leq -\frac{1}{4}\beta\frac{\|g_s\|_2}{2\frac{B\Gamma^2}{\mu_2^4(\mathcal{L})}\sqrt{n}\max_i\{\eta_i\}} \leq$$

$$-\frac{1}{8}\beta\frac{1}{\frac{B\Gamma^2}{\mu_2^4(\mathcal{L})}\sqrt{n}} = -\frac{\beta}{8\sqrt{n}p\delta}\frac{\gamma^3}{\Gamma^2}\frac{\mu_2^4(\mathcal{L})}{\mu_n^3(\mathcal{L})}$$

*The the quadratic decrease phase we use the result of Lemma 3 and induction:*

1. *For $m = 1$ applying $\alpha_s = 1$ in Equation (22):*

$$\|g_{s+1}\|_2 \leq \frac{B\Gamma^2}{\mu_2^4(\mathcal{L})}\|g_s\|_2^2 + \hat{B} \leq \frac{1}{4\frac{B\Gamma^2}{\mu_2^4(\mathcal{L})}} + \hat{B}$$

   *This result validates the claim for $m = 1$.*

2. *Let us assume it is correct for some $m > 0$.*

3. *Using $\alpha_{s+m+1} = 1$ in Equation (22) and denoting $u = 2^{2^m}$ gives :*

$$\frac{B\Gamma^2}{\mu_2^4(\mathcal{L})}\|\boldsymbol{g}_{s+m+1}\|_2 \le \left[\frac{B\Gamma^2}{\mu_2^4(\mathcal{L})}\|\boldsymbol{g}_{s+m}\|_2\right]^2 + \frac{B\Gamma^2}{\mu_2^4(\mathcal{L})}\hat{B} \le$$

$$\left[\frac{1}{u} + \hat{B}\frac{B\Gamma^2}{\mu_2^4(\mathcal{L})} + \tilde{\Lambda}\frac{\frac{1}{2}u-1}{u}\right]^2 + \frac{B\Gamma^2}{\mu_2^4(\mathcal{L})}\hat{B} =$$

$$\frac{1}{u^2} + \frac{B\Gamma^2}{\mu_2^4(\mathcal{L})}\hat{B} + \tilde{\Lambda}\frac{\frac{1}{2}u^2-1}{u^2} - \tilde{\Lambda}\frac{\frac{1}{2}u^2-1}{u^2} + \tilde{\Lambda}\frac{u-2}{u^2} + \hat{B}\frac{2B\Gamma^2}{\mu_2^4(\mathcal{L})}\frac{1}{u} +$$

$$\left(\frac{B\Gamma^2}{\mu_2^4(\mathcal{L})}\right)^2\left[\hat{B}^2 + 2\hat{B}\tilde{\Lambda}\frac{1}{\frac{B\Gamma^2}{\mu_2^4(\mathcal{L})}}\frac{(u-2)}{u} + \tilde{\Lambda}^2\frac{1}{\left(\frac{B\Gamma^2}{\mu_2^4(\mathcal{L})}\right)^2}\frac{(u-2)^2}{4u^2}\right]$$

*Since $\hat{B} + \frac{B\Gamma^2}{\mu_2^4(\mathcal{L})}\hat{B}^2 = \frac{\tilde{\Lambda}}{4\frac{B\Gamma^2}{\mu_2^4(\mathcal{L})}}$, then*

$$\frac{B\Gamma^2}{\mu_2^4(\mathcal{L})}\|\boldsymbol{g}_{s+m+1}\|_2 \le$$

$$\frac{1}{u^2} + \frac{B\Gamma^2}{\mu_2^4(\mathcal{L})}\hat{B} + \tilde{\Lambda}\frac{\frac{1}{2}u^2-1}{u^2} - \tilde{\Lambda}\frac{\frac{1}{2}u^2-1}{u^2} + \tilde{\Lambda}\frac{u-2}{u^2} + \hat{B}\frac{2B\Gamma^2}{\mu_2^4(\mathcal{L})}\frac{1}{u} +$$

$$\left(\frac{B\Gamma^2}{\mu_2^4(\mathcal{L})}\right)^2\left[\tilde{\Lambda}\frac{1}{4\left(\frac{B\Gamma^2}{\mu_2^4(\mathcal{L})}\right)^2} - \frac{\hat{B}}{\frac{B\Gamma^2}{\mu_2^4(\mathcal{L})}} + \frac{2\hat{B}\tilde{\Lambda}}{\frac{B\Gamma^2}{\mu_2^4(\mathcal{L})}}\left(\frac{1}{2} - \frac{1}{u}\right) + \frac{\tilde{\Lambda}^2}{\left(\frac{B\Gamma^2}{\mu_2^4(\mathcal{L})}\right)^2}\frac{(u-2)^2}{u^2}\right] =$$

$$\frac{1}{u^2} + \frac{B\Gamma^2}{\mu_2^4(\mathcal{L})}\hat{B} + \tilde{\Lambda}\frac{(u^2-2)}{2u^2} + \frac{\tilde{\Lambda}}{u^2}\left[-\frac{1}{2}u^2 + u - 1\right] + \frac{\tilde{\Lambda}}{4} + \hat{B}\frac{B\Gamma^2}{\mu_2^4(\mathcal{L})}\frac{2}{u} +$$

$$\hat{B}\tilde{\Lambda}\frac{B\Gamma^2}{\mu_2^4(\mathcal{L})}\left[1 - \frac{2}{u}\right] + \tilde{\Lambda}^2\left(\frac{u-2}{2u}\right)^2 = \frac{1}{u^2} + \frac{B\Gamma^2}{\mu_2^4(\mathcal{L})}\hat{B} + \tilde{\Lambda}\frac{(u^2-2)}{2u^2} -$$

$$\frac{\tilde{\Lambda}}{u^2}\left(\frac{u}{2}-1\right)^2 + \tilde{\Lambda}^2\left(\frac{1}{2} - \frac{1}{u}\right)^2 + \hat{B}\frac{B\Gamma^2}{\mu_2^4(\mathcal{L})}\left[-1 + \frac{2}{u} + \tilde{\Lambda} - \frac{2}{u}\tilde{\Lambda}\right] =$$

$$\frac{1}{u^2} + \frac{B\Gamma^2}{\mu_2^4(\mathcal{L})}\hat{B} + \tilde{\Lambda}\frac{(u^2-2)}{2u^2} - \left(\frac{1}{2} - \frac{1}{u}\right)^2\left(\tilde{\Lambda} - \tilde{\Lambda}^2\right) - \hat{B}\frac{B\Gamma^2}{\mu_2^4(\mathcal{L})}(1-\tilde{\Lambda})\left(1 - \frac{2}{u}\right) \le$$

$$\frac{1}{u^2} + \frac{B\Gamma^2}{\mu_2^4(\mathcal{L})}\hat{B} + \tilde{\Lambda}\frac{(u^2-2)}{2u^2} = \frac{1}{2^{2^{m+1}}} + \hat{B}\frac{B\Gamma^2}{\mu_2^4(\mathcal{L})} + \tilde{\Lambda}\left[\frac{2^{2^{m+1}-1}-1}{2^{2^{m+1}}}\right]$$

*The last step follows due to $u > 2$ and $\hat{\Lambda} < 1$ (choosing $\epsilon$ small enough).*

*Hence, our claim is correct.*

## Footnotes

[1]In this case $n = p = dk$