[Reviews · NeurIPS 2018]

Reviewer 1



In this paper, the authors studied the problem of multitask reinforcement learning (MTRL), and propose several optimization techniques to alleviate the scalability issues observed in other methods, especially when the number of tasks or trajectories is large. Specifically, they rely on consensus algorithms to scale up MTRL algorithms and avoid the issues that exist in centralized solution methods. Furthermore, they show how MTRL algorithms can be improved over state-of-the-art benchmarks by considering the problem from a variational inference perspective, and then propose a novel distributed solver for MTRL with quadratic convergence guarantees. In general, this work is tackling some important problems in the increasingly popular domain of multi-task RL. Using the variational perspective of RL, the problem of MTRL can be cast as a variational inference problem, and policy search can be done through the minimization of the ELBO loss. To alternate the updates on variational parameters and the policy parameters, the authors also propose using EM based approaches, which is very reasonable. Furthermore, by imposing the policy parameterization (as a combination of shared weights and separate weights) that is intuitive for multi-task learning, they showed that the policy search algorithm can be solved in a distributed way, and proposed using Laplacian consensus algorithm (instead of ADMM) to improve convergence In general, I found this paper well-written and the concepts are clearly described. I also find the derivations and the convergence analysis of the consensus algorithms rigorous. I also appreciate the detailed comparison between the proposed method and other state-of-the-art Newton-based method. My only question here is the necessity of using the variational formulation. It seems that the paper is built based on two separate parts, one part on deriving the ELBO loss, and the second part on developing consensus algorithms for MTRL. It seems that with the above parameterization one can still derive consensus algorithms for original MDP objective functions and argue using policy gradient type arguments. In this case is calculating Hessian going to be more difficult? It would be great to understand (or state in the paper) why is it necessary to use the variational formulation. Another minor comment: I believe the Chebyshev polynomial is only used in the parameterization of the Hessian H(\lambda), can the authors define these notations just before section 4.2?

Reviewer 2



Summary: This paper presents a distributed Multitask Reinforcement Learning (MTRL) from a variational inference perspective. It improves the state-of-art ADMM by a distributed solver using Chebyshev polynomials with \textit{quadratic convergence guarantees}. The theoretical analysis is verified with extensive experiments, but to strengthen the result, further experiments on real large-scale Multitask Reinforcement Learning problems are needed. Quality: Technically, this paper is rather strong. First, the MTRL is formulated as a variational inference problem by introducing a variational trajectory distribution to arrive at the maximization of variational lower bound. Variational distribution and policy parameters (shared and task-specific) could be optimized alternatively. Furthermore, to learn the shared parameter, the authors incorporate Laplacian matrices from Graph structure into the objective function, with a consensus constraint for the shared parameter to share and transfer knowledge. To solve the scalability issues, the authors adopt Chebyshev polynomials and de-centralize, distribute the computations of Newton direction, and prove the quadratic convergence rate. The experiments' scales are pretty small, even for the larger random network with only 150 nodes and 250 edges. It is better to study the really large-scale problems and investigate the speedup w.r.t computation units e.g. the number of CPU/GPU. Clarity: The majority of this paper is clear and well written and organized. However, there are some minor issues. -- Notation inconsistency: for action, line 57, 58, 67 use $a$, while line 103 uses $u$; for $q$, line 208 denotes dual function $q(\lambda)$, yet line 76 denotes variational distribution $q_\phi(\tau)$. -- Figures: Fig. 2(a) is too small, it might be better to use log-log plot for Fig. 6 -- Algorithm: it might be better to move algorithm 1 into the main paper, instead of the Appendix. -- Experiment: As communication overhead is not ignorable, it is better to show the comparison of overall time (communication + running time). -- Computing Architecture: It is better to illustrate how master and worker nodes are organized to distribute the Newton direction computation. This will help us understand the distribution algorithm and communication overhead. -- Results: In Fig. 3(b) for CP, SDD-Newton does not seem to be the best. In Fig. 6(c), distributed SDD Newton is no better than Distributed ADMM for the final objective values. Originality: The major contribution of this paper is to adopt Chebyshev polynomials and distribute the computing of Newton search direction with a proof of quadratic convergence rate. It is natural to use Chebyshev polynomials to solve the system of linear equations described via diagonally dominant matrices, where the Newton search direction is such kind of problem. However, the distributed Chebyshev Solver is non-trivial and it is an interesting idea to introduce such a technique into distributed Multitask Reinforcement Learning. Significance: Scalability is an important issue in Multitask Reinforcement Learning and it is the right direction to distribute the computation to as many machines as possible. Thus, the quadratic convergence guarantees an important result for the distributed Multitask Reinforcement Learning. However, the trade-off between communication and computation is the next important issue needed to be seriously considered. ------- After Rebuttal --------- I have checked the others' reviews. Most of my concerns are addressed by the author feedback.

Reviewer 3



This paper introduces a new algorithm for solving multitask RL problems. The claim of the paper is that by introducing a new solver, the authors were able to derive a quadratically convergent algorithm. The contribution of the paper is that it introduces a solver for optimizing the 'shared parameters'. There seems to be a extensive background literature that is related to the current paper that I am not familiar with at all. The authors, however, makes extensive references to these previous approaches. I also found the premise of the paper very narrow. I don't think the proposed approach is directly applicable to complicated domains. (For example, I don't think the proposed approach can help solving problems on Atari games where one has to solve multiple games at once.) I could be greatly misunderstanding the paper and therefore I would like to ask the following questions. Questions: 1. How is the policy parameterized? Is it a Gaussian policy? 2. Does the algorithm have access to the state transition probabilities during the optimization process? 3. Does the algorithm have access to all states of the MDP or does it have to conduct exploration? 4. What is jump start versus asymptotic performance in the experiment section? 5. Why do we care about quadratic convergence at all when it comes to multi-task RL? 6. What are some of the applications of this multi-task RL setup?